

# Snapshots of mean ocean temperature over the last 700,000 yr using noble gases in the EPICA Dome C ice core

Marcel Haeberli[1], Daniel Baggenstos[1], Jochen Schmitt[1], Markus Grimmer[1], Adrien Michel[1,2], Thomas Kellerhals[1], and Hubertus Fischer[1]

[1]Climate and Environmental Physics & Oeschger Centre for Climate Change Research, University of Bern, Switzerland,Sidlerstrasse 5, CH-3012 Bern
[2]Laboratoire des Sciences Cryosphérique, EPFL, ENAC IIE CRYOS GR A0 455 (Bâtiment GR), Station 2, CH-1015 Lausanne

**Correspondence:** hubertus.fischer@climate.unibe.ch

**Abstract.**

Together with the latent heat stored in glacial ice sheets the ocean heat uptake carries the lion's share of glacial/interglacial changes in the planetary heat content but little direct information on the global mean ocean temperature (MOT) is available to constrain the ocean temperature response to glacial/interglacial climate perturbations. Using ratios of noble gases and molecular nitrogen trapped in the Antarctic EPICA Dome C ice core we are able to reconstruct MOT for peak glacial and interglacial conditions during the last 700'000 years and explore the differences between these extrema. To this end, we have to correct the noble gas ratios for gas transport effects in the firn column and gas loss fractionation processes of the samples after ice core retrieval using the full elemental matrix of $N_2$, Ar, Kr and Xe in the ice and their individual isotopic ratios. The reconstructed MOT in peak glacials is consistently about 3.3 ± 0.4 °C cooler compared to the Holocene. Lukewarm interglacials before the Mid Brunhes event 450 kyr ago are characterized by 1.6 ± 0.4 °C lower temperatures in the ocean than the Holocene, thus, glacial/interglacial amplitudes were only about 50% of those after the Mid Brunhes event, in line with the reduced radiative forcing by lower greenhouse gas concentrations and their Earth system feedbacks. Moreover, we find significantly increased MOTs at the onset of Marine Isotope Stage 5.5 and 9.3, which are coeval with $CO_2$ and $CH_4$ overshoots at that time. We link these $CO_2$ and $CH_4$ overshoots to a resumption of the Atlantic Meridional Overturning Circulation, which is also the starting point of the release of heat previously accumulated in the ocean during times of reduced overturning.

## 1 Introduction

Over the last million years, Earth's climate has experienced pronounced changes in global ice volume (Bintanja et al., 2005), and hence sea level, accompanied by significant temperature changes between cold glacials and warmer interglacials. These temperature changes are observed with different amplitudes both on land (EPICA community members, 2004; Tzedakis et al.,





2006; Melles et al., 2012) as well as in the ocean (Elderfield et al., 2012; Shakun et al., 2012, 2015). In particular, due to the large size and heat capacity of the ocean reservoir, global mean ocean temperature (MOT) changes (together with the latent heat stored in waxing ice sheets) represent an integrated signal of Earth's energy imbalance in the past (Baggenstos et al., 2019). For example, today the global ocean is taking up about 90% of the excess heat from anthropogenic global warming,

dominating the current changes in the planetary energy budget (Gleckler et al., 2016; von Schuckmann et al., 2020).

In contrast to today's warming, where radiative forcing is caused through the anthropogenic emissions of $CO_2$ into the atmosphere, past climate cycles are generally believed to be driven by orbital changes in the latitudinal and seasonal distribution of the incoming solar radiation (Milankovic, 1941) with changing greenhouse gas concentrations representing an important amplifying feedback. Only recently, Tzedakis et al. (2017) were able to establish a simple rule for the occurrence of full

deglaciations based on the amount of the effective incoming energy flux (defined as the caloric summer half year insolation at 65° N corrected by a term that takes into account the time elapsed since the previous deglacation). While this rule is in agreement with the occurrence of all full deglaciations over the last million years, it does not predict differences between individual interglacials in ice volume and climate. For example, Marine Isotope Stage (MIS) 11 is characterized by a very long interval of interglacial conditions and sea-level at least equally high as in MIS 5.5 and likely more than 6 m higher than in the

Holocene (Dutton et al., 2015) . However, its effective energy is just reaching the threshold suggested by Tzedakis et al. (2017), while MIS 5.5 exhibits the largest effective energy over the last 2.5 Myr. Moreover, lukewarm interglacials (i.e., interglacials significantly colder than the Holocene with increased ice volume) were observed in local temperature reconstructions and global sea level (EPICA community members, 2004; PAGES Past Interglacials Working Group, 2016; Shakun et al., 2015; Elderfield et al., 2012) in the time interval 800-450 kyr BP (before present, where present is defined as 1950). Again, the

effective energy during these lukewarm interglacials is not systematically lower than for the subsequent interglacials, requiring additional energetic or radiative changes in the climate system to explain the difference between the individual interglacials. MOT may play an important role in explaining these features, as a higher ocean heat content could represent a compensating mechanism for the excess energy not used for the melting of the larger remnant ice sheets during lukewarm interglacials. Vice versa, if lower MOT parallels larger ice volume during lukewarm interglacials, changes in the radiative balance of the Earth

are required to explain the differences between the interglacials before and after 450 kyr BP.

Quantitative estimates of the integrated heat content of the entire ocean are required to answer these questions. However, obtaining a representative estimate of the whole ocean heat content represents a formidable challenge (Hoffman et al., 2017; Shakun et al., 2015). Marine sediments are available for many but not all ocean areas and record only local sea surface conditions or deep ocean temperatures at the coring site. Moreover, the proxy temperature information gained from marine

sediments may be affected by biological processes, by sea level changes, or are limited in precision especially for cold deep ocean temperatures. Finally, obtaining globally representative MOT from individual marine archives requires the compilation of a globally representative set of marine sediment cores, including rigorous cross-dating of all the records, which is difficult for the pre-[14]C period.

The measurement of atmospheric noble gases trapped in Antarctic ice provides a unique opportunity to reconstruct changes

in MOT independently from marine records (Bereiter et al., 2018b; Headly and Severinghaus, 2007; Ritz et al., 2011) and may



overcome these limitations. Changes in the atmospheric concentrations of xenon, krypton and nitrogen (where we generously use the expression noble gas throughout this paper also for $N_2$, as its atmospheric residence time amounts to tens of millions of years) mirror the anomalies in total ocean heat content, as they are driven by changes in their temperature-dependent physical solubilities in ocean water. This allows us to reconstruct MOT in the past from a single ice core sample. Note that the MOT at

a given time in the past represents a snapshot of the heat (and noble gas) content of all water parcels of the ocean and, thus, integrates over water bodies with different ventilation ages. MOT is therefore not directly linked to mean global sea surface temperature at the same time.

Here we extend previous efforts in reconstructing MOT using noble gases in ice cores (Bereiter et al., 2018b; Baggenstos et al., 2019; Shackleton et al., 2020), which were limited to the last and penultimate glacial terminations, by reconstructing

the glacial/interglacial MOT range for snapshots of all glacial and interglacial intervals over the last 700,000 years using the EPICA Dome C (EDC) ice core. To obtain quantitative MOT values, large corrections of the raw data for physical transport processes in the firn have to be applied, which affect the precision and accuracy of the MOT data. Moreover, gas fractionation processes in the ice between bubbles and clathrates in connection with potential post-coring gas loss have been observed, which have to be quantified and corrected for to allow us to derive an unbiased MOT estimate, as described in detail in this study.

Despite these corrections, we are able to derive quantitative MOT values from high-precision measurements noble gas mixing ratios from the EDC ice core.

## 2 Measurement and corrections

### 2.1 Determination of noble gas elemental and isotopic ratios in ice cores

The data set presented here consists of 88 EDC ice samples (39 samples span the last 40 kyr (Baggenstos et al., 2019), 49

samples are from MIS 5 to MIS 17) that were analyzed at the University of Bern. In order to test the representativeness of our EDC MOT data, 4 samples from the EPICA Dronning Maud Land (EDML) ice core, 4 samples from the Greenland Ice Core Project (GRIP) ice core, 4 samples from the Greenland NEEM ice core, 4 samples from the Antarctic Talos Dome ice core and 13 blue ice samples from Taylor Glacier were measured as well. The Taylor Glacier samples stem from a lab intercomparison with the Scripps Institution of Oceanography in San Diego, which showed excellent agreement within the

analytical uncertainties between both labs. Two samples from EDC (age > 700 kyr BP) were discarded due to bad ice quality together with drill fluid contamination and one sample was rejected due to a procedural problem.

The gas extraction and processing follow broadly the method described in detail by Bereiter et al. (2018a). In short, after shaving off the outer surface ~600 grams of ice are melt-extracted (equivalent to ~55 ml STP of extracted air) and the released air is quantitatively frozen into stainless steel dip tubes in a helium exchange cryostat. To equilibrate the sample and to avoid

fractionation by thermal gradients in the dip tube, the dip tubes are kept for at least 10 hours in a ventilated isothermal box. The samples are then split into two aliquots. The larger (~53 ml STP) aliquot is exposed to a Zr/Al getter alloy at 900 °C to remove all reactive gases and is analyzed in a peak-jumping routine on a Thermo-Finnigan MAT 253 dual inlet isotope ratio mass spectrometer for the isotopic ratios of xenon, krypton and argon as well as their elemental ratios. The smaller aliquot



(∼2 ml STP) is passed through a $CO_2$ trap and measured on a Thermo-Finnigan MAT Delta V dual inlet isotope ratio mass spectrometer in parallel for argon, oxygen and nitrogen isotopes and their elemental ratios. The measurements were corrected for pressure imbalance and chemical slope according to the procedure described in Severinghaus et al. (2003). All data are reported in the delta notation with respect to a modern atmosphere standard collected outside the lab in Bern, Switzerland.

5   Note that this atmospheric value is reflecting the current dissolution equilibrium of the studied gases between the atmosphere and the global ocean, where this atmospheric standard is not subject to fractionation processes in the firn that affect air samples from ice cores. Accordingly, in the absence of any long-term changes in the total abundance of noble gas isotopes in the atmosphere and ocean, each determined ice core value provides a measure of the difference of the ocean heat content at the age of the sample relative to today.

**Table 1.** Long-term reproducibility of the analyses (one standard deviation) derived from 51 measurements of outside air on the MAT 253 and 46 measurements of outside air on the Delta V Plus.

| Ratio | Uncert. ($1\sigma$) | IRMS |
|:---|:---:|:---:|
| $\delta^{136/129}$Xe | 0.037‰ | MAT 253 |
| $\delta^{132/129}$Xe | 0.033‰ | MAT 253 |
| $\delta^{86/84}$Kr | 0.014‰ | MAT 253 |
| $\delta^{86/82}$Kr | 0.021‰ | MAT 253 |
| $\delta^{40/36}$Ar | 0.006‰ | MAT 253 |
| $\delta^{15}$N | 0.007‰ | Delta V Plus |
| $\delta^{132}$Xe/$^{84}$Kr | 0.148‰ | MAT 253 |
| $\delta^{129}$Xe/$^{84}$Kr | 0.156‰ | MAT 253 |
| $\delta^{132}$Xe/$^{40}$Ar | 0.197‰ | MAT 253 |
| $\delta^{129}$Xe/$^{40}$Ar | 0.173‰ | MAT 253 |
| $\delta^{84}$Kr/$^{40}$Ar | 0.125‰ | MAT 253 |
| $\delta^{40}$Ar/$^{28}$N$_2$ | 0.052‰ | Delta V Plus |
| $\delta^{32}$O$_2$/$^{28}$N$_2$ | 1.117‰ | Delta V Plus |

10   The long-term reproducibility of our system, which was determined using outside air samples, may be found in Table 1. Note that the uncertainties of the isotopic ratios for each gas are generally increasing with mass difference between the respective isotopes. When normalized to one unit of mass difference, all isotopic ratios can be quantified to better than 10 permeg with uncertainties as low as 5 permeg per difference mass unit for $\delta^{136/129}$Xe, 7 permeg per difference mass unit for $\delta^{86/84}$Kr and as low as 1.5 permeg per difference mass unit for $\delta^{40/36}$Ar. Only $\delta^{15}$N values measured in this study did not yet

15   reach the expected precision per unit mass difference. Accordingly, we refrain from using $\delta^{15}$N to correct for gravitational or thermodiffusion fraction in this study.

Despite this high analytical long-term reproducibility derived from outside air samples, some species showed a much higher scatter in samples derived from ice cores due to remnants of drill fluid in the gas sample after processing. Drill fluid interference



in the mass spectrometer alters $\delta^{15}$N and $\delta^{40/36}$Ar in the analysis of the smaller, non-gettered aliquot, but also $\delta^{86/82}$Kr in the gettered aliquot. The affected samples can be clearly identified by the large deviation from the majority of the samples and be discarded. Nitrogen isotopes were precisely measured by Dreyfus et al. (2010), Landais et al. (2013), and Bréant et al. (2019) in previous studies. Despite the lower precision of our $\delta^{15}$N analyses, our results for those samples not affected by drill fluid

5   contamination are on average in good agreement with these values (see also Fig. 3). Nevertheless, we refrain from using the $\delta^{15}$N and $\delta^{40/36}$Ar values measured using the smaller aliquot and $\delta^{86/82}$Kr values of the larger aliquot, because due to the drill fluid contamination isotope data for these species are not available for the complete data set.

To derive MOT, we need to know the changes in the noble gas mixing ratios ($\delta$Kr/N$_2$, $\delta$Xe/N$_2$, and $\delta$Xe/Kr). To obtain them, we mathematically combine $\delta$Xe/Ar, $\delta$Kr/Ar and $\delta$Ar/N$_2$. Ar itself is not used as reference element because argon is

10  preferentially excluded relative to N$_2$, xenon and krypton during the bubble formation at the firn ice transition (Severinghaus and Battle, 2006). The uncertainties calculated from the analytical uncertainties of the elemental ratios used in this calculation are 0.136‰ and 0.181‰ for $\delta$Kr/N$_2$ and $\delta$Xe/N$_2$, respectively.

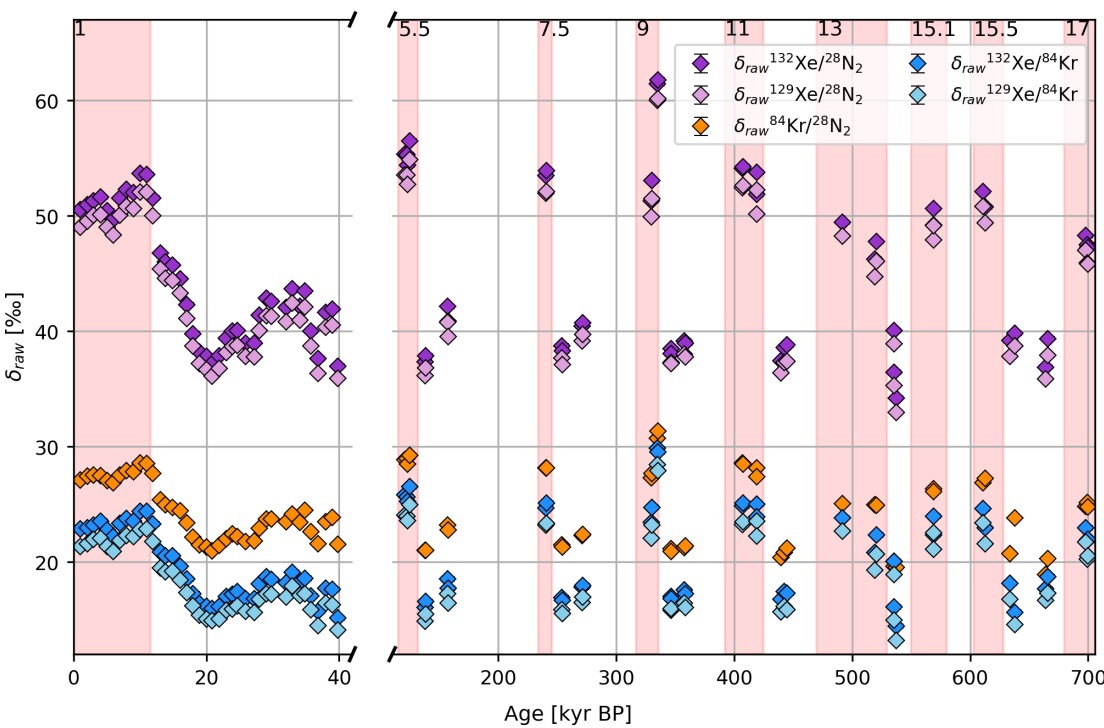

**Figure 1.** Measured delta values of the EDC elemental ratios from 700 kyr BP to present. The analytical uncertainties of the elemental ratios are too small (on the order of 0.1‰, see text) to be displayed here. The red shaded intervals indicate interglacials as identified by Masson-Delmotte et al. (2010).



## 2.2 Inferring atmospheric noble gas ratios from raw data

While the troposphere is well-mixed through turbulent processes, the low permeability of the firn restricts bulk flow, thus, the gas transport is controlled by molecular diffusion. All measured isotope and elemental ratios we extract from the ice core samples (reported as 'raw ratios') are, therefore, highly fractionated with respect to the atmospheric values (as illustrated by the strongly positive $\delta$ values in Figure 1), mainly due to gravitational enrichment in the diffusive zone of the firn but also by thermal diffusion if a temperature gradient is present in the firn column (Schwander et al., 1988), or by non-diffusive transport processes that disturb the diffusive equilibrium (so called kinetic fractionation). These systematic fractionations in the isotopic and elemental ratios have to be corrected for as illustrated in Figure 2.

Moreover, post-coring gas losses accompanied by gas fractionation processes have been observed in our samples in the Bubble Clathrate Transition Zone (BCTZ) and in poorly preserved and cracked ice mainly in the deepest section of the ice core. While the gas fractionation processes mentioned above can be corrected for using the full matrix of noble gases and their isotopes, the gas loss fractionation cannot yet be corrected for and affected samples have to be unambiguously identified using clear detection criteria and the respective values have to be removed from the data set. In the following sections, the systematic correction of the isotopic and elemental ratios is described, while the postcoring gas loss and its detection is described in chapter 3.

### 2.2.1 Systematic processes altering the isotopic and elemental ratios

#### Geological outgassing of $^{40}$Ar

In contrast to the nitrogen, krypton and xenon isotopes as well as $^{36}$Ar and $^{38}$Ar, the atmospheric abundance of $^{40}$Ar has gradually increased over time. This atmospheric $^{40}$Ar increase is caused by the integrated rate of $^{40}$Ar outgassing throughout Earth's history due to the radioactive decay of $^{40}$K in the crust and mantle (Bender et al., 2008). Assuming the $^{40}$Ar increase is constant in time, the effect can be corrected if the age of the sample is known. In this study we use $\delta^{40/36}$Ar values corrected for outgassing according to (Bender et al., 2008)

$$\Delta\delta^{40/36}\mathrm{Ar}_{\mathrm{outgas}} = 6.6 \cdot 10^{-5} \cdot t_{\mathrm{gas}} \tag{1}$$

Here the gas age $t_{\mathrm{gas}}$ has units of kyr BP. The outgassing effect on MOT is small for samples from the last transition (<0.05°C), but becomes more important for samples in the deeper and older ice (on the order of 1°C for the oldest samples).

#### Gravitational enrichment

Gas transport in the pore space of the firn column is controlled by molecular diffusion below a convective zone in the top meters of the firn. In this diffusive column molecules of different masses are separated leading to gravitational enrichment of the heavier gases and isotopes at the bottom of the firn column where bubble close-off occurs. Thus, all elemental and isotopic



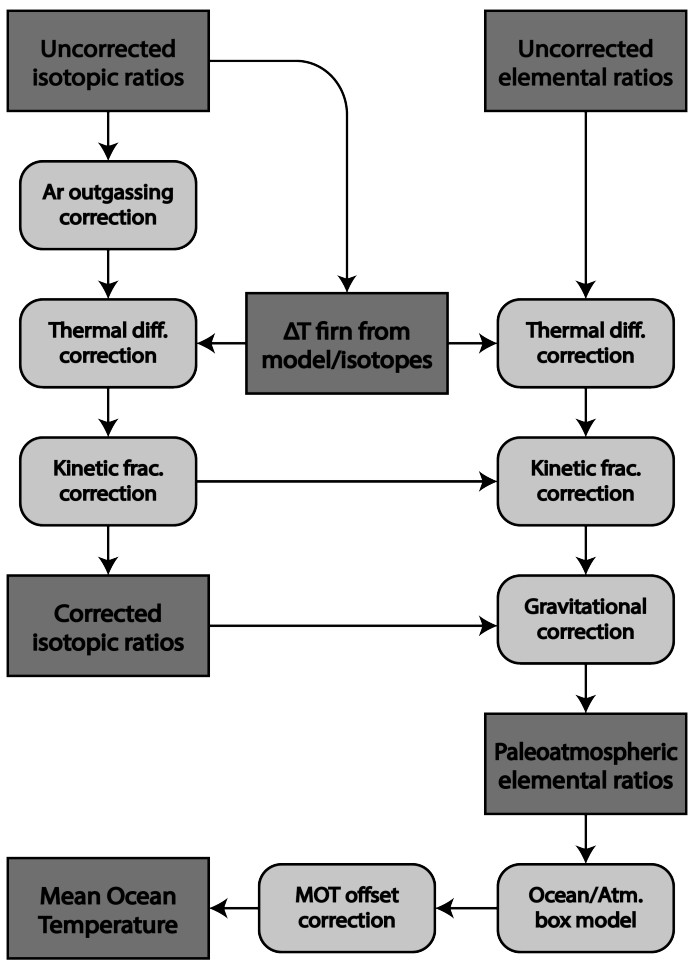

**Figure 2.** Flow scheme to infer paleoatmospheric elemental ratios and finally MOT using the measured elemental ($\delta Kr/N_2$, $\delta Xe/N_2$ and $\delta Xe/Kr$) and isotopic ratios ($\delta^{40/36}Ar$, $\delta^{86/84}Kr$, $\delta^{136/129}Xe$). Values are indicated by dark grey square boxes, while applied corrections are illustrated by rounded boxes in lighter grey. In a first step, Ar isotopes have to be corrected for changes in $^{40}Ar$ due to geological outgassing. The isotopic and elemental ratios are then corrected for thermal diffusion and kinetic fractionation. The temperature difference between the top and the bottom of the diffusive column height required for the correction of thermal diffusion is derived using either a firn column model (Michel, 2016) or the full matrix of noble gas isotopes. For the kinetic fractionation, we assume fixed ratios between isotopic and elemental ratios (Birner et al., 2018). Finally, the corrected isotopic ratios are used to correct the elemental ratios for the gravitational enrichment. The corrected elemental ratios are then translated into MOT using an ocean atmosphere box model. In a last step, the resulting MOT are corrected for their Holocene offset (see text for details).

ratios are strongly enriched in the heavier gases compared to the atmosphere according to

$$\Delta\delta_{\text{grav}}(z) = \delta_{\text{grav}}(z) - \delta_{\text{atm}} = ((\delta_{\text{atm}} + 1) \cdot e^{\frac{\Delta m\, g\, z}{R\, T}} - 1) - \delta_{\text{atm}} \qquad (2)$$



where $g$ is the standard gravitational acceleration, $R$ the gas constant, $T$ the mean firn temperature (Craig et al., 1988; Schwander, 1996). To correct for the gravitational enrichment of the elemental ratios at the lock-in depth we have to know the diffusive column height (DCH) $z$ of the firn (where $z$ is defined positive downward), which can be directly derived from the isotopic enrichment of the individual gas species after corrections for thermal diffusion and kinetic effects (see below).

## 5 Thermal diffusion

The second diffusive process leading to fractionation of the air in the firn column is thermal diffusion. It refers to the fractionation of a gas mixture in the presence of a temperature gradient. This leads to an enrichment of the heavier components at the cold end of the firn column according to the laboratory-determined thermal diffusion sensitivity $\Omega^{X/Y}$ for two gas or isotope species X and Y, where $\Omega^{X/Y}$ is specific for the gas mixture (Severinghaus et al., 2003):

$$\Delta\delta_{\text{therm}} = \Omega^{X/Y} \cdot \Delta T \tag{3}$$

Here $\Delta T$ is defined as the temperature difference between the top and the bottom of the DCH. As EPICA Dome C is a low accumulation site, the mean annual firn temperature increases with depth, i.e., the difference of the mean annual temperature between the surface and bottom of the firn column $\Delta T$ is always negative even in the absence of temporal climate changes at the surface (Ritz et al., 1982) and gas enclosed in bubbles is expected to be slightly depleted by thermal diffusion relative to

its gravitational value. Note that this thermal fractionation is gas species-dependent and much larger for the lighter gases. Here we use the thermal diffusion sensitivities given by Kawamura et al. (2013) and Headly (2008).

### Kinetic fractionation: the Heavy Isotope Deficit

If only diffusive processes occured in the firn column, the differences between any of the measured isotope ratios could be used to unambiguously reconstruct the thermal and gravitational fractionation components using the well-known thermal diffusion

sensitivity parameters (Kawamura et al., 2013; Headly, 2008). In particular, as the gravitational enrichment of two gases only differs according to the mass difference between two gases, the gravitational enrichment normalized to unit mass difference should be the same for all gases. However, even after the thermal diffusion correction the isotopic ratios of different gases per unit mass difference still reveal systematic offsets (see Fig. 3a), indicating that there is another mechanism at play that we have yet to correct for.

The reason for the systematic differences in the isotopic ratios per unit mass difference between the different gases at Dome C in Figure 3 is the occurrence of advective processes in the firn such as turbulent mixing at the surface (Kawamura et al., 2013), barometric pumping and the net vertical movement of air by ice flow and compression (Buizert and Severinghaus, 2016; Birner et al., 2018) that all disturb the diffusive equilibrium in the firn column. The term coined for this difference of measured isotopic ratios per unit mass difference after thermal correction is Heavy Isotope Deficit (HID) (Buizert and Severinghaus,

2016) or differential kinetic fractionation (Birner et al., 2018) between the isotopic or elemental ratio X and another isotopic or elemental ratio Y. In the following we will use the difference between the kinetic fractionation of an isotopic or elemental ratio X per unit mass difference relative to the kinetic fractionation in $N_2$ isotopes (Kawamura et al., 2013; Birner et al., 2018;





Buizert and Severinghaus, 2016). For example, for Ar isotopes this fractionation is defined as:

$$\epsilon^{*}_{\text{Ar}-\text{N}_2} = \Delta\delta^{40/36}\text{Ar}^{*}_{\text{kin}} - \Delta\delta^{15}\text{N}^{*}_{\text{kin}} \tag{4}$$

Here and in the following "*" indicates that the values are normalized to unit mass difference, for example $\delta^{40/36}\text{Ar}^{*}_{\text{kin}} = \delta^{40/36}\text{Ar}_{\text{kin}}/4$. Note that the heavier (and slowly diffusing) gases like xenon and krypton will be further from diffusive equi-
librium than lighter (and faster diffusing) gases like argon and nitrogen. Another way to put this is that the effective DCH of different gases is not the same, with $z$ being a little shorter for heavier, less diffusive gases.

In our measurements the total deficit of xenon and krypton isotopes relative to argon isotopes per unit mass difference over the last 40 kyr is on average -0.046 ± 0.007‰ and -0.029 ± 0.007‰ (here and throughout the manuscript all uncertainties refer to 1 sigma). This total deficit comes about by the combined effect of different thermal diffusion sensitivities of the different
gases and by gas specific kinetic fractionation. Separation of these two effects on the total deficit requires corrections for the thermal gradient as described in section 2.2.2. Note that due to the higher thermal sensitivity of Ar compared to Kr and Xe and the negative firn temperature difference at Dome C, the thermal diffusion effect increases the heavy isotope deficit compared to what is expected from kinetic fractionation only.

Birner et al. (2018) discovered that while the absolute magnitude of kinetic fractionation $\Delta\delta^{*}_{\text{kin}}$ may vary substantially be-
tween different ice core sites, the kinetic fractionations between two gases occur in fixed ratios between isotopic and elemental ratios for a variety of firn regimes. Thus, although we do not know the absolute kinetic fractionation at Dome C for an individual gas for past firn conditions a priori, we can use the results by Birner et al. (2018) together with our multiple gas species and transient firn temperature modeling to correct for the HID and other gas transport effects. It is worth noting that the ratios of the mean kinetic fractionations over the last 40 kyr as displayed in Figure 3 agree well within uncertainties with the ratios predicted
by Birner et al. (2018). However, for individual samples especially older than 40 kyr, the ratios can deviate considerably from the theoretical values, partly due to the analytical uncertainties but potentially also due to an isotope fractionating Ar loss that may have occurred during transport of the samples from Antarctica to Bern, where some samples warmed to temperatures above -10°C but stayed always well below the melting point. In the following we therefore used the mean kinetic fractionation over the last 40 kyr as representative value for the entire record and included the uncertainty of this mean value in our error
estimate.

**Gas loss effects on isotopic ratios**

As mentioned in the previous paragraph, ice core samples with heavy post-coring gas loss exhibit enriched $\delta^{40/36}\text{Ar}$, while $\delta^{15}\text{N}$ is unaffected (J. Severinghaus, personal communication). Due to their large size, the krypton and xenon isotopes are also not thought to be affected. Correction coefficients for argon isotope enrichment in bubble ice have been suggested by Kobashi
et al. (2008b), but it is not certain, whether these correction factors describe the gas loss quantitatively and in particular whether they can be used for ice from any depth and for all ice cores. We refrain from trying to correct for this effect because we do not see evidence for artifactual gas loss (see chapter 3.1) and because no correction coefficients have been established for clathrate ice.



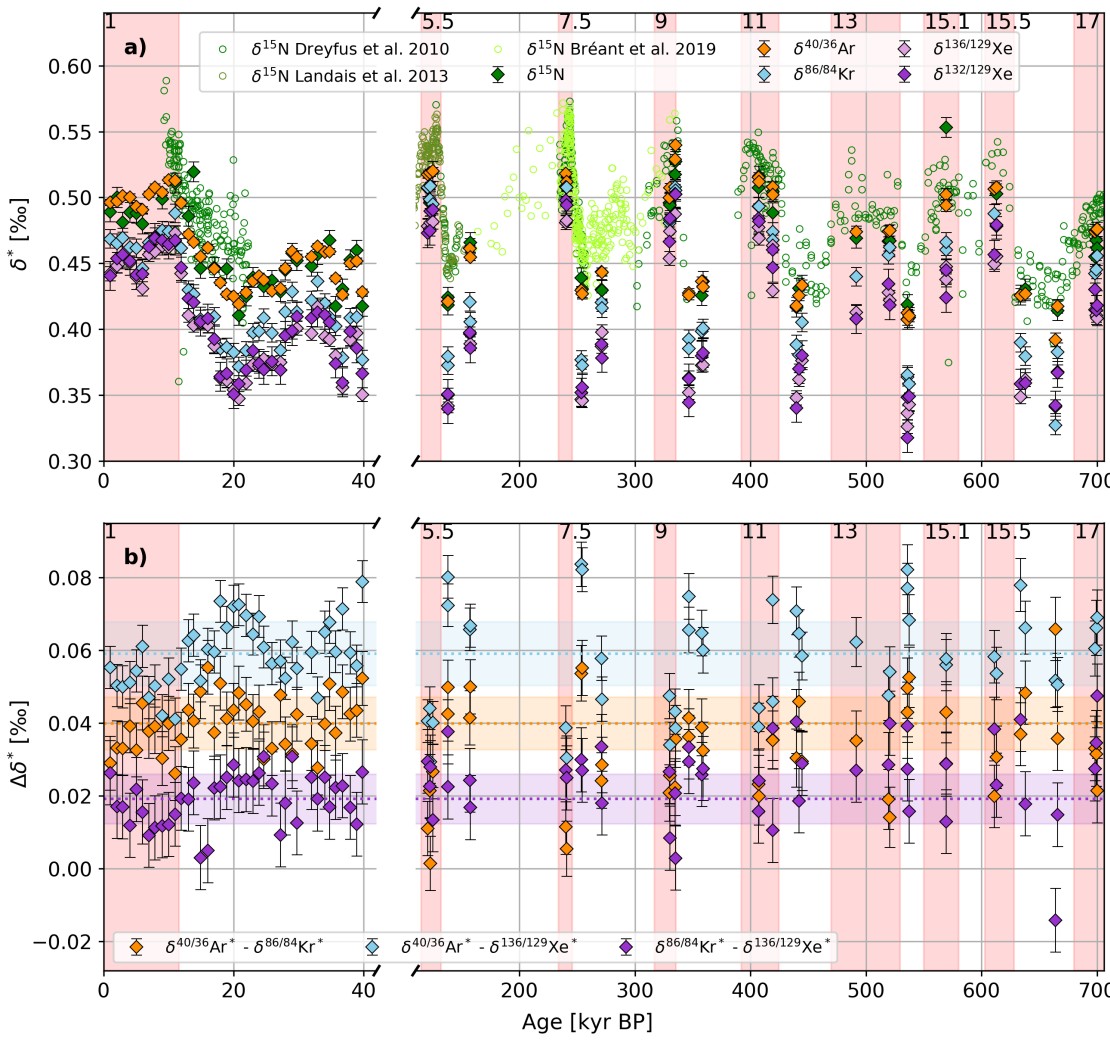

**Figure 3. a)** Isotopic enrichment of the EDC samples from 700 kyr BP to present. All isotopic data are shown per unit mass difference and reported relative to modern air sampled in Bern, Switzerland. The data is corrected for thermal fractionation as described in the text. In the case of $\delta^{15}$N, data points contaminated with drilling fluid have been excluded. The circles in darker green, olive, and lighter green represent measurements from the EDC ice core from Dreyfus et al. (2010), Landais et al. (2013), and Bréant et al. (2019), which have been corrected for thermal fractionation using our model-derived firn temperatures. **b)** Difference between the isotopic ratios of the EDC samples after correction for thermal fractionation. The dashed line is the mean difference from the samples over the last 40kyr with the shaded areas representing their standard deviation. The red shaded intervals and the MIS numbers on top indicate interglacials as identified by Masson-Delmotte et al. (2010).



### 2.2.2 Correcting isotopic ratios for transport processes in the firn column

As outlined above, the measured isotopic ratio $\delta^*_{\text{meas}}$ is influenced by a series of processes that alter the paleo-atmospheric value $\delta^*_{\text{atm}}$. We can write for the measured isotopic ratio:

$$\delta^*_{\text{meas}} = \delta^*_{\text{atm}} + \Delta\delta^*_{\text{grav}} + \Delta\delta^*_{\text{therm}} + \Delta\delta^*_{\text{kin}} \tag{5}$$

Note that we assume ocean solubility effects on the isotopic ratios to be negligible (in contrast to the elemental ratios, where we use these effects to quantify MOT). This can be justified using a two-box model calculation with the recently determined solubility fractionation factors in seawater (Seltzer et al., 2019) and imposed ocean temperature differences, which yields glacial/interglacial changes in atmospheric noble gas isotope composition of less than 0.001‰ per unit mass difference. We can, thus, safely assume that for isotopic ratios $\delta^*_{\text{atm}} = 0$ (where for Ar isotopes the atmospheric value has to be corrected for
Ar outgassing as described in 2.2.1).

   Using the measured isotopic ratios we are able to quantify the fractionation effects and correct the isotopic and elemental ratios. Here we explore two different correction pathways that differ in the derivation of $\Delta T$, which is required to quantify the thermal diffusion effect. Both possibilities use the matrix of isotopic ratios to constrain the gravitational enrichment and the HID. However, one possibility uses a 1D ice flow model connected to a dynamic heat advection and diffusion model (Michel,
2016) to quantify the thermal diffusion effect, the other uses the full matrix of noble gas isotopes measured in this study to quantify $\Delta T$ empirically.

### a) correction using a firn temperature model

In order to correct for the strong gravitational enrichment (large mass difference) we can use the precisely measured isotopic ratios after correcting for thermal diffusion and kinetic fractionation. Due to the negative firn temperature difference at EDC
and due to the negative kinetic fractionation for heavier gases, the corrected isotopic ratios are slightly increased compared to the measured ratio. The results for the thermally corrected isotopic ratios reflecting the remnant kinetic fractionation are shown in Figure 3a, where $\Delta T$ used for the thermal correction was calculated using an ice flow/heat flow model as described below. The isotopic ratio least affected by kinetic fractionation is $\delta^{15}\text{N}$, however, our $\delta^{15}\text{N}$ measurements are not as precise as $\delta^{40/36}\text{Ar}$ per unit mass difference and some of the $\delta^{15}\text{N}$ values had to be discarded due to drill fluid contamination affecting
the mass spectrometric analysis. Accordingly, we use $\delta^{40/36}\text{Ar}$ as our reference isotope. In many applications to reconstruct temperature using isotopic ratios of permanent gases (e.g. Kobashi et al. (2008a)) the small kinetic fraction of Ar relative to $\text{N}_2$ is neglected and the gravitational enrichment per unit mass difference $\Delta\delta^*_{\text{grav}}$ in units of permille is calculated according to

$$\Delta\delta^*_{\text{grav}} = \delta^{40/36}\text{Ar}^* - \Omega^{40/36*} \cdot \Delta T \tag{6}$$

where the "*" in $\Omega^{40/36*}$ indicates that the thermal sensitivity has been normalized to unit mass difference.
However, firn gas pumping and modeling experiments (Kawamura et al., 2013) show that all gases are subject to kinetic fractionation and therefore even Ar isotopes show a small heavy isotope deficit $\epsilon^*_{\text{Ar}-\text{N}_2}$ after correction of thermal diffusion





effects (Buizert and Severinghaus, 2016) compared to nitrogen isotopes. Thus, $\Delta\delta^*_{\text{grav}}$ should be calculated according to:

$$\Delta\delta^*_{\text{grav}} = \delta^{40/36}\text{Ar}^* - \Omega^{40/36*} \cdot \Delta T - \Delta\delta^{40/36}\text{Ar}^*_{\text{kin}} = \delta^{40/36}\text{Ar}^* - \Omega^{40/36*} \cdot \Delta T - \epsilon^*_{\text{Ar}-\text{N}_2} - \Delta\delta^{29/28}\text{N}^*_{2,\text{kin}} \tag{7}$$

The absolute kinetic fractionation $\Delta\delta^{40/36}\text{Ar}^*_{\text{kin}}$ in equation 7 is not known a priori and dependent on firn and meteorological conditions at the site that may have changed in the past. Using the equivalent equation for Xe isotopes and the linear relationship of the kinetic fractionations given by Birner et al. (2018), $\epsilon^*_{\text{Ar}-\text{N}_2}$ can be calculated from our measurements according to:

$$\epsilon^*_{\text{Ar}-\text{N}_2} = (\delta^{136/129}\text{Xe}^*_{\text{meas}} - \delta^{40/36}\text{Ar}^*_{\text{meas}} - (\Omega^{136/129*} - \Omega^{40/36*}) \cdot \Delta T)/6.3 \tag{8}$$

Note that while including the kinetic fractionation may improve the accuracy of the reconstruction, it decreases slightly the precision, as also the analytical error for Xe isotopes comes into play.

$\Delta T$ in equations 7 and 8 is unknown but can be estimated using an ice flow/heat flow model that solves the ice advection/firn compaction and the heat advection and diffusion equation transiently over the last million years. Here we used the model by Michel (2016) and a parameterization of heat diffusion and heat conductivity according to Cuffey and Paterson (2010). Density of the firn was estimated by the Herron Langway model (Michel, 2016). Heat diffusivity of the very porous top 10 m of the firn was corrected according to Weller and Schwerdtfeger (1970). The recent firn temperature difference $\Delta T$ in this run agreed within $\pm 0.2°$ C between the model and measured firn temperature profiles at Dome C (Buizert et al., 2020, submitted). These $\pm 0.2°$ C are in the following assumed to represent the analytical uncertainty of the model. Surface temperature and snow accumulation used in the model are prescribed using the EDC temperature record over the last 800,000 yr (Jouzel et al., 2007, Bazin et al., 2013) and a scaled version of the stacked marine $\delta^{18}\text{O}$ record (Lisiecki and Raymo, 2005) for earlier times. For an optimal fit with the measured temperature profile over the entire ice thickness at Dome C a $+2°$ C correction of glacial temperatures was applied by Michel (2016), however, this has only a very small effect on the firn temperature difference. Ice thickness changes over the last 800 kyr are taken from Parrenin et al. (2007) and a scaled version of the stacked marine $\delta^{18}\text{O}$ record for earlier times. The constant geothermal heat flux as well as the thinning function were chosen to optimize the agreement of the modeled temperature profile with borehole temperature measurements (Buizert et al., 2020, submitted) and the age profile at Dome C (Bazin et al., 2013).

The DCH affects both the gravitational enrichment and the absolute value of the temperature gradient. Accordingly, we use an iterative approach for equation 7 to determine the firn temperature gradient from the model results. In the first step we neglect the temperature gradient, thus thermodiffusion effects, to estimate the DCH

$$z = \ln\left(\frac{\Delta\delta^*_{\text{grav}}}{1000} + 1\right) \cdot \frac{\text{R}T}{\Delta m^* \text{g}} \tag{9}$$

where $\Delta m^*$ is the unit mass difference in kg/mol. After this first step we read out the modeled firn temperature difference for this depth and use it for the thermal diffusion correction in the next step. Using this iterative approach the firn column depth already converges after two iterations. This model-based analysis shows that the firn column depth is about 70-80 m for glacial periods and 90-100 m for interglacials.

The kinetic fractionation $\epsilon^*_{\text{Ar}-\text{N}_2}$ calculated using modeled $\Delta T$ is about -0.009±0.001 ‰ (1$\sigma$) over the last 40 kyr. Due to analytical uncertainty or a potential isotope fractionating Ar loss during transport of the samples, the values of $\epsilon^*_{\text{Ar}-\text{N}_2}$ for



individual samples older than 40 kyr significantly deviate from the mean value over the last 40 kyr (see above). We, therefore, use the mean $\epsilon^*_{Ar-N_2}$ value over the last 40 kyr for the entire record and include its standard deviation as a measure of a potential systematic error that we introduce by this choice of $\epsilon^*_{Ar-N_2}$.

The last unknown in equation 7 and 4 is the amount of kinetic fractionation for nitrogen isotopes $\Delta\delta^{15}N^*_{kin}$. Modeling and firn air studies by Buizert and Severinghaus (2016) and Birner et al. (2018) show that at WAIS divide, i.e., a site on the West Antarctic plateau, this fractionation is less than 0.005 ‰, while at Law Dome, a site subject to strong barometric pumping, it can be as high as 0.050 ‰. Assuming a kinetic fractionation at Dome C that is similar to WAIS, this small fractionation in $\Delta\delta^{15}N^*_{kin}$ translates into an offset in the final MOT values on the order of 0.01°C and can, therefore, be safely neglected in our MOT reconstruction. In the following, we assume $\Delta\delta^{15}N^*_{kin} = 0$ for Dome C, where synoptic pressure variations and wind speeds are rather low. Neglecting potential errors in the thermal diffusion sensitivities, the uncertainty in $\Delta\delta^*_{grav}$ introduced by this model-based method to correct the data is 0.003 ‰ (1$\sigma$).

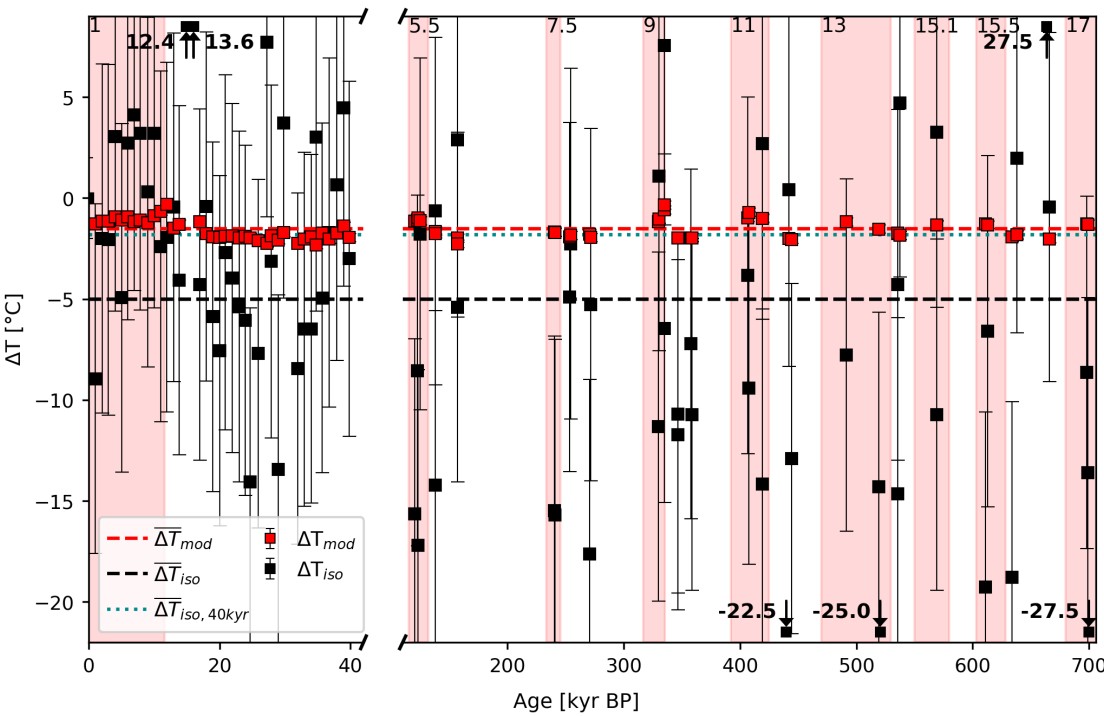

**Figure 4.** Comparison of the firn temperature difference $\Delta T_{mod}$ (top-bottom) as derived from the ice flow-heat flow model of Michel (2016) (red squares), and $\Delta T_{iso}$ obtained using measured isotopic ratios only (black squares). Dashed lines in red and black indicate the overall mean values for $\Delta T_{mod}$ and $\Delta T_{iso}$, while the dotted line in cyan represents the mean $\Delta T_{iso}$ when considering samples from the first 40kyr only. The red shaded intervals and the MIS numbers on top indicate interglacials as identified by Masson-Delmotte et al. (2010).





## b) correction using measured isotopic ratios only

Using the firn temperature model as in a) assumes that the firn temperature difference in the model over the DCH realistically resembles the temperature difference seen by the gas. The model neglects for example seasonal variations in the temperature gradient, which may not cancel out in the long-term annual average, and also assumes that the current parameterizations used

for firn density, heat conduction and heat diffusivity profiles in the firn are also representative for past conditions. An alternative way is, therefore, to use only experimental evidence from our full matrix of isotope ratios. Assuming that gravitational enrichment, thermal diffusion and kinetic fractionation completely describe the fractionation processes occurring in the firn column, this approach provides a measurement-based estimate of $\epsilon^*_{\mathrm{Ar-N_2}}$ and $\Delta T$.

The equivalent to equation 8 for Kr isotopes is

$$\epsilon^*_{\mathrm{Ar-N_2}} = (\delta^{86/84}\mathrm{Kr}^*_{\mathrm{meas}} - \delta^{40/36}\mathrm{Ar}^*_{\mathrm{meas}} - (\Omega^{86/84*} - \Omega^{40/36*}) \cdot \Delta T)/4.25 \tag{10}$$

with the constant ratio of kinetic fractionations for different isotopes given by Birner et al. (2018).

Solving equation 8 for $\Delta T$ and inserting it into equation 10, it follows that

$$\epsilon^*_{\mathrm{Ar-N_2}} = \frac{(\delta^{86/84}\mathrm{Kr}^* - \delta^{40/36}\mathrm{Ar}^*) - \alpha(\delta^{136/129}\mathrm{Xe}^* - \delta^{40/36}\mathrm{Ar}^*)}{4.25 - 6.3\alpha} \tag{11}$$

where $\alpha = (\Omega^{86/84*} - \Omega^{40/36*})/(\Omega^{136/129*} - \Omega^{40/36*})$. Note that the kinetic fractionation $\epsilon^*_{\mathrm{Ar-N_2}}$ derived using this data based

approach is on average -0.009 ‰ over the last 40 kyr and, thus, in perfect agreement with the results from the firn temperature model approach.

Using equation 11 in equation 8 finally leads to

$$\Delta T = \frac{(\delta^{136/129}\mathrm{Xe}^* - \delta^{40/36}\mathrm{Ar}^*) - \frac{6.3}{4.25 - 6.3\alpha}((\delta^{86/84}\mathrm{Kr}^* - \delta^{40/36}\mathrm{Ar}^*) - \alpha(\delta^{136/129}\mathrm{Xe}^* - \delta^{40/36}\mathrm{Ar}^*))}{\Omega^{136/129*} - \Omega^{40/36*}} \tag{12}$$

While this method in principle avoids any potential systematic uncertainties in the firn temperature gradient, it substantially

increases the uncertainty of the reconstruction, as now the analytical uncertainties of all three noble gas isotopic ratios have to be taken into account for each individual sample. Using Monte Carlo error propagation, the uncertainty in $\epsilon^*_{\mathrm{Ar-N_2}}$ derived in this approach is increased to 0.01 ‰ (i.e., roughly ten times larger than in the model-based approach) and the uncertainty in $\Delta T$ is about 8°C. This translates into an error for $\Delta\delta^*_{\mathrm{grav}}$ of roughly 0.08 ‰ (i.e., almost 30 times higher than in the model-based approach) and finally into an error in the corrected elemental ratios (see section 2.2.3) of several permille, which is prohibitive

for a reliable reconstruction of MOT from individual samples.

The very high scatter in data-derived $\Delta T$ is displayed in Fig. 4. Despite the high scatter there may be some systematic variability over the last 40 kyr, however it is not directly correlated to the surface temperature evolution. Also, the range of $\Delta T$ values in the Holocene is the same as in the glacial period. At first order, we, therefore, assume a normal distribution of $\Delta T$ values over the last 40 kyr in this approach. The standard deviation of $\Delta T$ over the last 40 kyr is 5.8°C, which is of the same

order as the expected analytical uncertainty of 8°C derived from error propagation. Hence, this large scatter can be readily explained by analytical noise.





Looking at Figure 4 it should be stressed that despite the large scatter the mean $\Delta T$ over the last 40 kyr derived in this data-based approach is in excellent agreement with $\Delta T$ derived in the model-based approach. This points to the fact that using our noble gas isotope measurements we are able to accurately (but not sufficiently precisely) reconstruct $\epsilon^*_{\text{Ar}-\text{N}_2}$ and $\Delta T$ over the last 40 kyr and that thermal diffusion and kinetic fractionation fully explain the systematic offsets of the isotopic ratios.

However, for the samples older than 40 kyr the average $\Delta T$ is -7.7 $\pm$ 9.7 °C, which is physically not in line with the observed or modeled firn temperature differences and also not in line with the results younger than 40 kyr. As we do not expect that any unknown firn fractionation processes were active prior to 40 kyr BP that did not occur over the last 40 kyr, this apparent offset in $\Delta T$ is either reflecting the large scatter of the data or is related to an isotope fractionating Ar loss process during transport, when the samples got relatively warm for a short period of time.

As mentioned above, using the individual data-based approach to correct the elemental ratios for each sample separately is not a viable solution, as the scatter in reconstructed $\Delta T$ and $\epsilon^*_{\text{Ar}-\text{N}_2}$ and following in MOT is too large. Therefore, we regard the mean $\epsilon^*_{\text{Ar}-\text{N}_2}$ and $\Delta T$ values over the last 40 kyr to be representative for the entire record and use those to calculate $\Delta\delta^*_{\text{grav}}$. To quantify the systematic uncertainty introduced by this choice of $\epsilon^*_{\text{Ar}-\text{N}_2}$ and $\Delta T$, we calculate an upper (lower) estimate of $\Delta T$ and $\epsilon^*_{\text{Ar}-\text{N}_2}$ as defined by their mean values over the last 40 kyr plus (minus) their standard deviation. This will provide us

with a range of values for $\Delta\delta^*_{\text{grav}}$ for the systematic corrections of the elemental ratios (see 2.2.3).

We stress again that while such a constant correction with a systematic range of possible values has a significant impact on the absolute reconstructed MOT value, the impact of the correction on the glacial/interglacial difference in MOT is small. Moreover, the glacial/interglacial difference in MOT is very similar whether one uses the model-based approach in a) or the mean data-based approach in b). Accordingly, only MOT differences relative to the Holocene are considered, as was also the

case in the study by Shackleton et al. (2020) .

### 2.2.3 Correction of the elemental ratios

Equivalent to isotopic ratios of one gas species, the ratios of two different gas species (here referred to as elemental ratios) have to be corrected for gravitational enrichment. Due to the large mass difference between two gas species, this gravitational correction of elemental ratios is substantial. In addition, thermal diffusion as well as kinetic fractionation have to be corrected

in the elemental ratios. For the Kr/N$_2$ ratio this can be summarized according to:

$$\delta\text{Kr}/\text{N}_{2,\text{atm}} = (\delta\text{Kr}/\text{N}_{2,\text{meas}} - \Omega^{\text{Kr}/\text{N}_2}\Delta T - (\epsilon^*_{\text{Kr}/\text{N}_2-\text{N}_2} + \Delta\delta^{15}\text{N}^*_{\text{kin}}) \cdot \Delta m/\Delta m^* + 1) \cdot (\Delta\delta^*_{\text{grav}} + 1)^{-\Delta m/\Delta m^*} - 1 \qquad (13)$$

where again we set $\Delta\delta^{15}\text{N}^*_{\text{kin}} = 0$, as the effect of $\Delta\delta^{15}\text{N}^*_{\text{kin}}$ is negligible compared to other uncertainties. The first factor in brackets on the right hand side represents the measured value of the elemental ratio corrected for thermal diffusion and kinetic fractionation, while the second factor corrects this isotopic ratio for its gravitational enrichment. $\Delta T$, $\epsilon^*_{\text{Kr}/\text{N}_2-\text{N}_2}$ and $\Delta\delta^*_{\text{grav}}$ can

be calculated from the firn temperature model and from the isotopic data as described above using the fixed ratios of kinetic fractionation given by Birner et al. (2018). Note that the exponent $\Delta m/\Delta m^*$ is very large (e.g. 56 for $^{84}\text{Kr}$ and $^{28}\text{N}_2$), thus, the gravitational correction is substantial and sensitive to any analytical error in the isotopic ratios as well as in uncertainties in the thermal correction and the kinetic fractionation.



Here we use both the model-based and the mean data-based reconstruction of $\Delta T$ and $\epsilon_{\mathrm{Kr/N_2-N_2}}$ described above and apply these to equation 13 and equivalent equations for Xe/N$_2$ and Xe/Kr ratios. The results of these corrections are shown in panel a (model-based reconstruction) and b (data-based reconstruction) of Fig. 5. Note that the whiskers in Fig. 5 represent the stochastic error ($1\sigma$) due to error propagation of the analytical uncertainties, while the size of the boxes represents the

systematic uncertainties ($1\sigma$) introduced by our choice of mean $\Delta T$ and $\epsilon_{\mathrm{Kr/N_2-N_2}}$. The first observation is that the three different atmospheric elemental ratios have different glacial/interglacial amplitudes, primarily reflecting the different temperature sensitivities of gas solubility in the ocean. Secondly, the corrected elemental ratios show that although our data are referenced to recent outside air at Bern, reflecting late Holocene ocean temperatures, the Holocene elemental ratios after all corrections are not equal to zero but systematically shifted to positive values.

Holocene elemental ratios larger than zero have also been recently observed by Shackleton et al. (2020) using Taylor Glacier ice, reflecting our insufficient quantitative understanding of fractionation processes in the firn, requiring more dedicated firn gas sampling and firn gas transport modeling studies that realistically resolve seasonal variations in the gas transport, pressure variability and kinetic fractionation. However, glacial/interglacial differences in our corrected elemental ratios are not sensitive to this overall offset. If we use changes in atmospheric elemental ratios relative to the Holocene mean, both the model-based

and the mean data-based approach provide very similar results. For example, using the model-based approach, the corrected last glacial $\delta$Xe/Kr values are on average about 2.9‰ lower than the Holocene values, which is in agreement within uncertainties with the results by Bereiter et al. (2018b) using ice from the WAIS divide ice core.

The final observation pertains the scatter of the data. Elemental ratios corrected using $\Delta T$ and $\epsilon_{\mathrm{Kr/N_2-N_2}}$ derived for each individual sample provide no meaningful results, as the analytical scatter introduced from all three isotopic ratios is too large.

Using the model-based approach, we introduce uncertainty through the elemental ratio analysis, the uncertainty in our model-based $\Delta T$ and systematic uncertainty through our use of the mean $\epsilon_{\mathrm{Kr/N_2-N_2}}$ over the last 40 kyr to correct the kinetic fractionation. This leads to a substantially smaller scatter than in the individual data based approach. The stochastic errors ($1\sigma$) in $\delta$Kr/N$_{2,\mathrm{atm}}$, $\delta$Xe/N$_{2,\mathrm{atm}}$ and $\delta$Xe/Kr$_{\mathrm{atm}}$ introduced by the measurement errors are 0.20‰, 0.45‰ and 0.33‰, respectively, in our model-based approach. The systematic uncertainty in this case is 0.23‰, 0.70‰ and 0.48‰, respectively.

Alternatively, we can use the mean data-based correction approach. Using the mean $\Delta T$ and $\epsilon_{\mathrm{Kr/N_2-N_2}}$ over the last 40 kyr leads to the smallest stochastic error in our atmospheric elemental ratios, as we average over 39 data points for our correction. However, this implies also a large systematic uncertainty that derives from the choice of mean $\Delta T$ and $\epsilon_{\mathrm{Kr/N_2-N_2}}$. Moreover both parameters may have been subject to systematic changes over time that we cannot decipher using this mean correction. The stochastic errors ($1\sigma$) in $\delta$Kr/N$_{2,\mathrm{atm}}$, $\delta$Xe/N$_{2,\mathrm{atm}}$ and $\delta$Xe/Kr$_{\mathrm{atm}}$ introduced by the measurement errors are 1.69‰, 2.14‰

and 0.67‰, respectively, in our mean data-based approach. The systematic uncertainty in this case is on average 1.22‰, 1.49‰ and 0.40‰, respectively.

In summary, our different correction pathways lead to similar results in the differences of atmospheric elemental ratios relative to their Holocene value. In the following, we will use only the atmospheric elemental ratios after correction using the model-based approach, which is based on well-understood physical laws of heat conduction/advection and provides elemental





**Figure 5.** Comparison of the reconstructed atmospheric ratios of Kr/N$_2$ (orange), Xe/N$_2$ (purple), and Xe/Kr (blue), **a)** as derived from the calibrated firn model of Michel (2016), **b)** using measured isotopic ratios only. The reconstructed value is drawn as a black line in the middle of each box. The magnitude of the systematic uncertainty is indicated by the size of the colored boxes and the statistical uncertainties are depicted as whiskers (1 $\sigma$ each). The red shaded intervals and the MIS numbers on top indicate interglacials as identified by Masson-Delmotte et al. (2010).

ratios with sufficient precision and accuracy to reconstruct glacial/interglacial changes in MOT after correction for gravitational, thermal and kinetic fractionation.





### 2.3 Box model to infer the MOT from atmospheric values

To reconstruct MOT values from the corrected elemental ratios, we use the ocean-atmosphere box model described by Bereiter et al. (2018b). The basic assumption in the model is that $N_2$, Kr and Xe are conserved in the ocean-atmosphere system and that these gases are in equilibrium between the two reservoirs. The model accounts for the effects of changes in ocean salinity, volume, and atmospheric pressure on the oceanic inventories of each of the three gases, which have individual temperature-dependent solubilities. The records of Lambeck et al. (2014) and Spratt and Lisiecki (2016) are used to reconstruct the sea level for the last 27 kyr and the time interval 28-800 kyr BP, respectively. In contrast to Bereiter et al. (2018b), we assume no change in the saturation state of the heavy noble gases in the ocean (Hamme and Severinghaus, 2007; Loose et al., 2016).

Using this box model we can translate the reconstructed and corrected atmospheric elemental ratios ($Kr/N_2$, $Xe/N_2$ and $Xe/Kr$) into changes in MOT relative to the Holocene. Using our model-based correction approach described above leads to a MOT change over the last glacial/interglacial transition of on average $2.9 \pm 0.4$ °C, $3.3 \pm 0.4$ °C and $3.6 \pm 0.4$ °C for $\delta Kr/N_{2,atm}$, $\delta Xe/N_{2,atm}$ and $\delta Xe/Kr_{atm}$, respectively (see Figure 6 and Table 2). This is in general agreement within uncertainties with independently derived results over the last glacial termination for the WAIS Divide ice core from Bereiter et al. (2018a). $\delta Xe/Kr_{atm}$ suggests somewhat colder glacial temperatures in our reconstruction, although postcoring gas loss might bias these data (see section 3.1).

**Table 2.** Absolute MOT anomaly based on $Kr/N_2$ in °C relative to the Holocene mean using both of our correction approaches. The Holocene value is the average of 10 EDC samples from 0-10.0 kyr BP, the LGM value is the average of 9 samples between 17.8-26 kyr BP, LIG is the average of 4 samples between 120-129.9 kyr BP, and MIS 6 includes 4 samples from 135-160 kyr BP. MIS 9 and MIS 10 each comprise 4 samples.

| Correction Method | HOL | LGM | LIG | MIS 6 | MIS 9 | MIS 10 | Stat. uncert. |
|---|---|---|---|---|---|---|---|
| Model | 0.0 | -2.9 | 0.8 | -3.2 | 0.7 | -3.5 | 0.4 (1$\sigma$) |
| Isotopes | 0.0 | -2.9 | 0.8 | -3.2 | 0.7 | -3.5 | 2.9 (1$\sigma$) |

As described above, the uncertainties in the model-based correction approach for elemental ratios are introduced by uncertainties in the modeled $\Delta T$, the measurement errors of the elemental ratios, and the choice of $\epsilon_{Kr/N_2-N_2}$. For the Ar isotopes we also corrected for the steady increase of $^{40}$Ar through the decay of $^{40}$K in the crust and mantle, which leads to a small long-term increase in $\delta^{40/36}$Ar. This increase was experimentally determined by Bender et al. (2008) and has a slope of 0.066 $\pm$ 0.007 ‰/My, which we include in our error propagation. Moreover, the box model approach may add some minor systematic uncertainty, which we cannot quantify. Using the analytical uncertainties in the elemental ratios in the box model approach leads to a stochastic uncertainty (1$\sigma$) of the MOT values between 0.34-0.39 °C for all three elemental ratios and a systematic uncertainty of 0.38-0.50 °C.



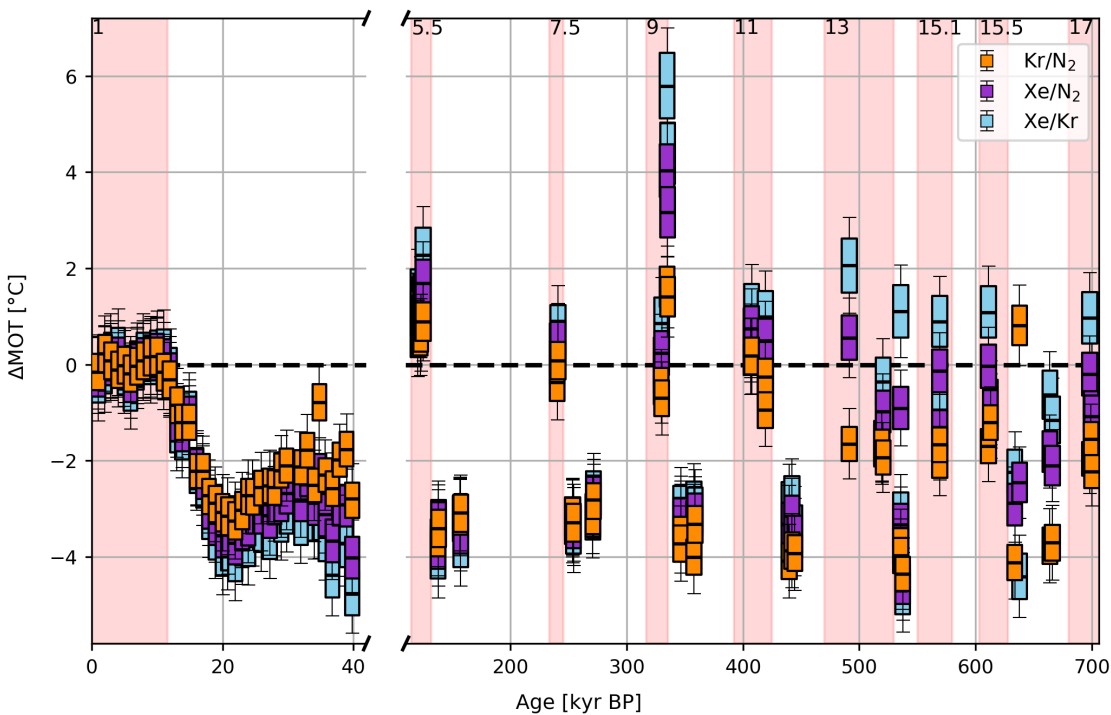

**Figure 6.** Reconstructed MOT for the EDC samples over the last 700 kyr relative to the Holocene value (using our model-based approach for the correction). Provided in orange are the MOTs derived from $\delta Kr/N_2$. Purple squares are MOTs using $\delta Xe/N_2$, and blue squares are MOTs based on $\delta Xe/Kr$. The horizontal line in the middle of the filled box provides the corrected value. The boundaries of the filled boxes refer to the systematic uncertainty, while whiskers reflect the stochastic uncertainty introduced by the measurement errors (each $1\sigma$). The red shaded intervals and the MIS numbers on top indicate interglacials as identified by Masson-Delmotte et al. (2010).

## 3 Postcoring gas loss effects on MOT reconstructions

Additional changes in the elemental ratios may occur if post-coring gas loss leads to fractionation between different gases. While visible cracks are avoided in the sample preparation, previous ice core studies showed that invisible microcracks can occur, leading to gas loss fractionation processes affecting for example $\delta O_2/N_2$ and $\delta Ar/N_2$, but little is known about gas loss

5   effects on heavy noble gas ratios. Ice subject to such internal microcracks is most susceptible to post-coring gas loss and in line we find systematic deviations in the MOT values derived from the three different elemental ratios in (i) ice from the Bubble Clathrate Transition Zone (BCTZ), which is characterized by brittle ice, and (ii) in the oldest, bottom-most meteoric ice of the EDC core, which is characterized by particularly large ice crystals and high in situ temperatures and which is prone to increased micro-fracturing. The following sections address potential gas loss fractionations in these ice zones.



### 3.1 Postcoring gas loss effects in the BCTZ

Post-coring fractionation of elemental gas ratios is caused by the selective loss of specific gases in poorly preserved samples or in samples from the Bubble to Clathrate Transition Zone (BCTZ), i.e., the depth section between the formation of the first clathrate and the disappearance of the last bubble (Lipenkov, 2000).

Bender (2002) called such gas loss through cracks or microcracks 'core-cracking fractionation' and observed this effect to strongly affect $O_2/N_2$ in the BCTZ of the Vostok ice core. This fractionation process is explained by gas fractionation occurring between co-existing bubbles and clathrates in combination with post-coring gas loss. In our record we observe small but linearly increasing differences with age (and depth) in the MOT derived from the different noble gas combinations in ice at the top of the BCTZ, i.e., in the zone where clathrates form (600-775 m; 24-40 kyr BP). In particular, MOT derived from $Kr/N_2$ show

higher temperatures compared to $Xe/N_2$ and $Xe/Kr$ in this ice, while the three values agree within their uncertainties in pure bubble ice. Similar systematic offsets are observed around the BCTZ in cores from various sites (GRIP, NEEM and TALOS) but to a variable extent as shown in Figure 7b. Bereiter et al. (2018a) also observed pronounced alteration in the trapped gas towards the lower end of the BCTZ or even below for the WAIS ice core.

    In the following we will try to motivate the processes that we think are responsible for this fractionation. Studies of air

bubbles and clathrate inclusions on the NGRIP ice core reveal two stages of clathrate formation: primary bubble clathratization and secondary clathrate growth (Kipfstuhl et al., 2001). Primary bubble clathratization takes place in the upper part of the BCTZ, where primary clathrates are formed directly from individual bubbles. The partial pressures of the individual gases in the bubbles increase linearly with depth due to the increasing hydrostatic pressure. In contrast, the partial pressures of the gases forming the primary clathrates is controlled by the dissociation pressure of the air clathrate, which is the dissociation pressure

of clathrates for the individual gas species multiplied by the atmospheric mole fraction of each individual gas. Note that only the $N_2$ partial pressure is high enough at the top of the BCTZ to instigate clathrate formation. Here we assume that most other gases are passively included in the $N_2$ sII clathrate structure (Sloan and Koh, 2007) but contribute to its stabilization. Deeper in the transition zone, secondary clathrate growth sets in, giving rise to the formation of large secondary clathrate crystals, where gases permeate from bubbles to coexisting clathrates within the BCTZ and where larger hydrates grow on the cost of smaller

ones (Ostwald ripening) below the BCTZ (Uchida et al., 2011).

    We attribute the gradually increasing offsets in reconstructed MOT at the top of the BCTZ to core-cracking gas loss in combination with the fractionation of gases during the second stage of clathratization. This fractionation effect requires the diffusive flux of air molecules with different permeation constants in ice from air bubbles to clathrate hydrates (Ikeda et al., 1999). As described above, the partial pressure of individual gases in the bubbles is larger than the effective dissociation

pressure of coexisting clathrates, driving permeation of gas molecules through the ice matrix from the bubbles to the clathrates. Because the permeation constants of Xe, Kr, $O_2$ and $N_2$ in ice are different, their diffusive fluxes to the clathrates are different; $O_2$ diffuses especially fast and is, thus, enriched in clathrates relative to $N_2$ (Ikeda et al., 2000) in the BCTZ, while the large and heavy, slow diffusing Xe gas is enriched in the bubbles. Moreover, we may speculate that Xe, which is per se a sI clathrate forming gas (Sloan and Koh, 2007), may not be as easily incorporated in the sII clathrate structure that is formed from bubbles





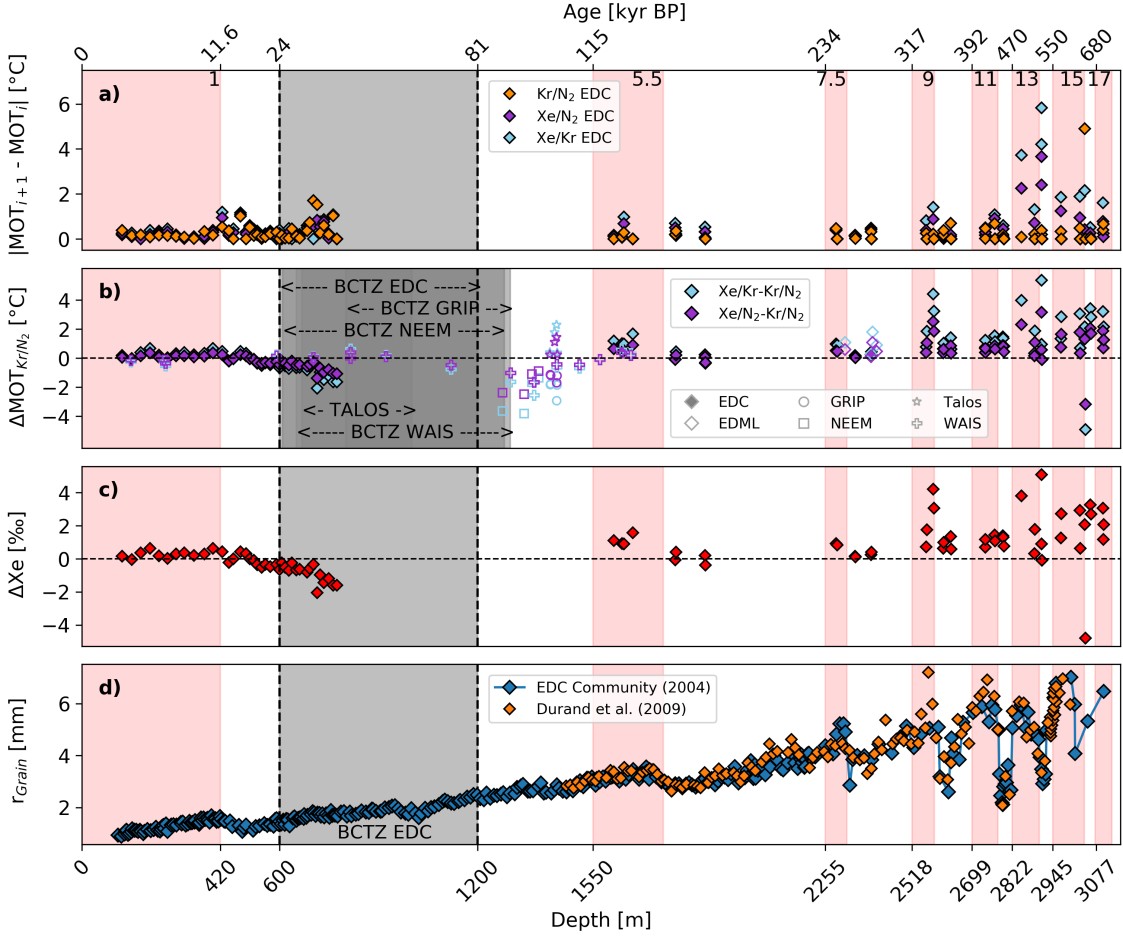

**Figure 7.** a) Absolute MOT pair difference of two adjacent samples for the three different atmospheric noble gas ratios. MIS interglacials (red shaded intervals) as identified by Masson-Delmotte et al. (2010) are labeled at the top for orientation. b) Difference of the MOT from $\delta Kr/Xe$ and $\delta Xe/N_2$ compared to $\delta Kr/N_2$. EDC, GRIP, NEEM and Talos samples were measured at the University of Bern. WAIS samples were measured by Bereiter et al. (2018b). The grey shaded bars and arrows indicate the BCTZ for the different cores as identified by Neff (2014). All data are corrected for gravitational and thermal fractionation using argon isotopes. EDC samples are also corrected for HID fractionation. c) Potential xenon change required for the EDC samples to match the MOTs using a least square approach. Given in red is the potential xenon change which minimizes the difference between the MOTs derived from Xe/Kr, Kr/N_2 and Xe/N_2. Negative values indicate a xenon loss, i.e., a depletion of the raw elemental ratios $Xe/N_2$ and Xe/Kr caused by postcoring gas loss. Positive values represent apparent net xenon 'gains' due to higher losses of krypton and nitrogen compared to xenon. d) evolution of the grain radius in the EDC core (EPICA community members, 2004; Durand et al., 2009).

in the ice. This would leave Xe enriched in the bubbles in the ice or dissolved in the ice matrix after complete disappearance





of the bubbles below about 1200 m. In any case, these processes would lead to a depletion of Xe relative to $N_2$ and Kr in the clathrate and an enrichment in remaining bubbles.

Gas enclosed in bubbles is more likely to be lost through cracks after core retrieval than gas in hydrates, thus, if post-coring gas loss from bubbles occurs, the remaining gas of the ice sample will be enriched in $O_2$ relative to $N_2$ and depleted in Xe. The
small but significant $\delta Xe/Kr$ and $\delta Xe/N_2$ decrease in the BCTZ can therefore be explained by the observation that during the gas loss process, Xe is preferentially lost via cracks from the Xe-enriched bubbles.

To quantify a potential Xe loss we use a least squares approach that artificially changes the Xe abundance in the ice to minimize the difference between the three MOT proxies. The respective calculated post-coring Xe loss is shown in Figure 7c, where the permil value is the change of the elemental ratio used to minimize the MOT differences compared to $Kr/N_2$. It
shows that Xe loss is almost negligible in the top 500 m, i.e., in bubble ice, and increases to about 2‰, meaning that the $Xe/N_2$ elemental ratio is depleted in Xe by up to 2‰ caused by post-coring Xe loss from ice in the BCTZ.

Previous studies have shown that the gas loss effect on $O_2/N_2$ is most pronounced when the ice is brittle with an enhanced number of cracks but that the quality of the data is also affected in non-brittle ice if the samples are stored for an extended amount of time at relatively warm temperatures of around -25 °C (Ikeda-Fukazawa et al., 2005; Landais et al., 2012). The
storage temperature may also affect our noble gas ratio results because our samples from the last transition were stored for about 15 years at about -22 °C. The samples from older climatic intervals presented in this study were stored at Dome C and later in Bern in a freezer at below -50 °C. Thus, no gas loss is expected for these samples during storage. However, these samples experienced a short-term warming during transport, when cooling stopped during transit. Temperatures were as warm as -16 °C at the bottom of the well-insulated transport box and -6 °C on top of the box for several hours, however the ice never
reached the melting point.

To assess the potential loss of gas due to high storage or transport temperature, we compare the $O_2/N_2$ that we measure alongside the noble gases with previous $O_2/N_2$ measurements on EDC ice stored at -50°C by Extier et al. (2018), where the latter had not experienced significant warming during transport. Similar to previous observations in BCTZ ice, our $O_2/N_2$ values suggest a variable enrichment of up to 4‰ in $O_2$ relative to $N_2$ at the top of the BCTZ and also slightly elevated $Ar/N_2$
relative to the values in pure bubble ice (as shown in Figure 8). This suggests that our samples from the BCTZ are indeed subject to slight core-cracking fractionation. For the deeper ice no significant offset relative to the $O_2/N_2$ data from Extier et al. (2018) can be found, suggesting that the short cooling interruption during transport did not affect our elemental ratios. For the $Ar/N_2$ ratios of samples older than 40 kyr BP we see a number of glacial values that are more depleted than the average value during the Last Glacial Maximum (LGM), however, the large spread of the data does not allow us to draw a firm conclusion on
an Ar loss for these older samples. Furthermore, both $O_2/N_2$ and $Ar/N_2$ of neighbouring samples agree well with each other, indicating a common cause for the large overall variability. The main factor influencing $O_2/N_2$ on these timescales are changes in local summer insolation (Kawamura et al., 2007; Extier et al., 2018), and $Ar/N_2$ should be affected in a similar fashion. This suggests that gas loss during transport and storage are of minor importance, as such artifactual gas loss is typically a chaotic process that randomly affects some samples more than others. Based on this, we refrain from making a gas loss correction for
$\delta^{40/36}Ar$ as described in section 2.2.1.



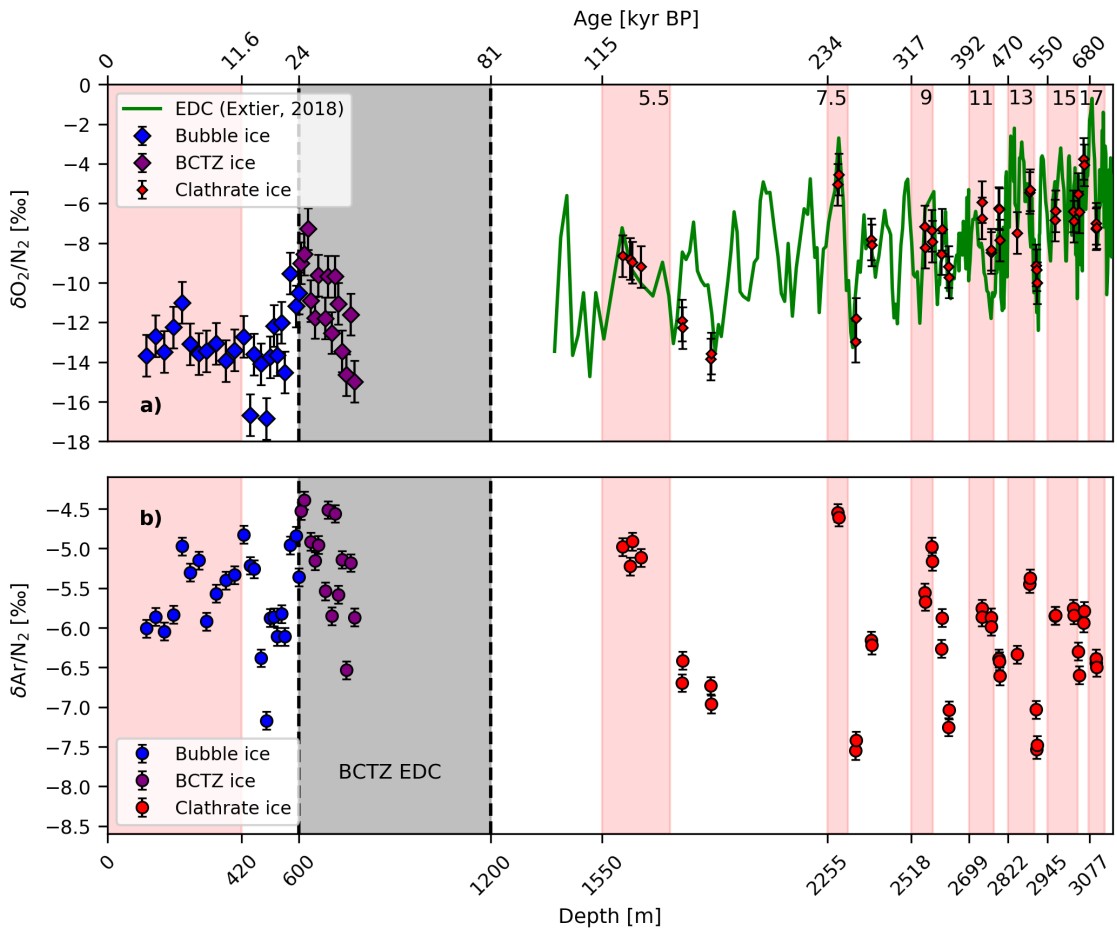

**Figure 8.** Measurements of $\delta O_2/N_2$ and $\delta Ar/N_2$ from the Dome C ice core on the EDC depth scale. The $\delta O_2/N_2$ record from Extier et al. (2018) is corrected for gravitational enrichment using $\delta^{15}N_2$. Our $\delta O_2/N_2$ data are corrected for gravitational enrichment using $\delta^{40/36}Ar$, while the $\delta Ar/N_2$ data are corrected for gravitational enrichment, thermal diffusion, kinetic fractionation, and geological Ar-outgassing. The data show a large variability in the BCTZ and reproducible values for clathrated ice. Interglacial MIS (red shaded areas) as identified by Masson-Delmotte et al. (2010) are labeled for orientation. a) Green line represent measurements from Extier et al. (2018), diamonds represent the EDC measurements of $\delta O_2/N_2$ of this study. b) EDC measurements of $\delta Ar/N_2$ from this study.

In summary, although the Xe depletion is relatively small in the ice at the top of the BCTZ, we believe that the MOTs derived from our Xe/Kr and Xe/N₂ values are slightly biased towards too cold temperatures. Accordingly, we use the MOT values derived from Kr/N₂ to quantitatively reconstruct changes in MOT for ice from the BCTZ, as Kr/N₂ appears not to be affected by gas loss and as also Kr/N₂ replicates agree well within their analytical uncertainty, thus, making an effect of
5   (unreproducible) gas loss unlikely. The latter also suggests that any fractionation of Kr relative to N₂ that may be caused by a permeation fractionation between bubbles and clathrates is too small to have a sizable effect on our measured values.



## 3.2 Post-coring gas loss in ice deeper than 2500 m

In clathrate ice below the BCTZ, MOT values derived from the three different elemental ratios showed good agreement down to an age of about 450 kyr BP, but revealed systematic offsets mainly for the lowest 400 m, where the MOTs derived from $\delta Xe/Kr$ and $\delta Xe/N_2$ suggest much higher temperatures than those derived from $\delta Kr/N_2$. A similar observation can be made for

two interglacial samples from MIS 9. Moreover, the values derived from $\delta Xe/Kr$ and $\delta Xe/N_2$ of neighboring, quasi-replicate samples in the lowest 400 m of the core do not agree within their analytical uncertainty, while those for $\delta Kr/N_2$ do, as shown by differences in MOT of each sample pair in Fig. 7a. Each of the $\delta Kr/N_2$ pairs (with the exception of one sample at about 3100 m depth (630 kyr BP)) agrees within less than 0.5°C, in line with our stochastic error estimate (see Figure 7). The cause for the outlier in $\delta Kr/N_2$ at 630 kyr BP has not yet been identified. Interestingly, $\delta Kr/N_2$ pair differences in the bottom-most ice

are also not elevated compared to shallower fully clathrated ice samples in the time interval $100 - 450$ kyr BP. Taking the good reproducibility of $\delta Kr/N_2$ and the elevated MOTs derived from $\delta Xe/Kr$ and $\delta Xe/N_2$ in the deepest ice together, we conclude that a non-fractionating Kr and $N_2$ core-cracking loss must have occurred, which leaves the ice enriched in Xe.

In the EDC core, Durand et al. (2009) observed large variations in the fabric of the ice in different samples and extraordinarily large grains in interglacial ice below 2500 m as shown in Figure 7. A recent microscopic inspection of two deep EDC ice

samples stored at the Alfred-Wegener-Institute in Bremerhaven at -30 °C showed significant visual glittering in this ice, indicative of so-called plate-like inclusions (PLI) due to the clathrate relaxation processes (Bereiter et al., 2015). These suggest the coexistence of reformed gaseous inclusions and clathrates in this increasingly relaxed ice. Weikusat et al. (2012) and Nedelcu et al. (2009) studied the ice structure and relaxation of the EDML core. Although the ice is comparatively young in this core and the accumulation rates are larger than at the EDC site, it is likely that the observations of EDML also apply to some degree

to the EDC core. Particularly interesting are relaxation-induced air inclusions such as microscopic bubbles ('microbubbles') and PLI that are found with increasing depth at grain boundaries and clathrate hydrate surfaces (Nedelcu et al., 2009). The PLIs are flat inclusions of air (a few micrometer in thickness) and are more common in deep, fully clathrated ice, while the microbubbles are spherical objects and more common in shallower, bubbly ice (Weikusat et al., 2012). Direct measurements of such microscopic air inclusions are only possible for the main air constituents $N_2$ and $O_2$ using Raman spectroscopy. The

results from Nedelcu et al. (2009) show substantially higher $O_2/N_2$ values in such inclusions compared to air, indicating an enrichment of $O_2$ in these gas-filled features similar in magnitude to the $O_2$ enrichment reported for air hydrates in the BCTZ of polar ice cores.

As diffusive gas loss from the outer surface of our samples can be safely ruled out due to the large sample size and the careful removal of several mm of ice from each surface (Bereiter et al., 2009) and due to the complete gas extraction achieved in our

melt extraction technique, any occurring net gas fractionation in our samples requires a gas loss process through microcracks and some gas fractionating exchange between gas enclosures in the ice. Similar to the core-cracking gas loss in the BCTZ discussed above, any gas loss through microcracks in the ice is more likely for gas filled features such as bubbles and PLIs than for clathrates. Hence, we would assume that a gas loss from $O_2$ enriched PLIs would lead to a depletion in the $O_2/N_2$ values measured in our ice samples. As shown in Fig. 8, however, an $O_2$ enrichment is observed that is increasing for older ice and





which has been attributed to a long-term change in the $O_2$ content of the atmosphere over the last 800 kyr (Stolper et al., 2016). Thus, either an $O_2$ loss did not occur or it is very small and dwarfed by the atmospheric $O_2$ change.

Nevertheless, we suggest that the apparent enrichment of Xe relative to $N_2$ and Kr in our data is related to a fractionation/gas loss process that is opposite to that what is observed in the BCTZ and which is related to the clathrate relaxation process. The

fractionation is apparently particularly pronounced in ice below 2500 m with large crystals (see Figure 7d) and experiencing in situ temperatures warmer than -10 °C, where this ice may experience enhanced micro-cracking during core retrieval and storage. Once the ice core is brought to the surface, it starts to relax and gas filled features such as PLIs slowly form (with the relaxation being the faster, the higher the storage temperature of the core). Note that due to the very cold storage temperature of our samples older than 40 kyr (< -50°C) at Concordia Station and later in a -50 °C freezer chest in our lab, relaxation of

the core is strongly suppressed and we expect much less PLIs to have formed in our samples compared ot the EDML samples in Nedelcu et al. (2009). Thus, it is likely that any $O_2$ loss is too small to be discerned. However, as discussed in section 3.1, the sII clathrates are likely to be depleted in Xe and part of the Xe is expected to remain dissolved in the ice matrix. Thus, while any gas inclusions reforming from clathrates will be depleted in Xe and can be subject to gas loss, the Xe molecules dissolved in the ice are not accessible to gas loss. Accordingly, we speculate that any gas loss leads to an enrichment of Xe

in our deep EDC ice samples, while all other gas species do not show a significant fractionation as long as very cold storage ensures the clathrate relaxation to be small. This is also the reason why Kr/$N_2$ ratios apparently do not show any signs of gas loss fractionation as illustrated by the small pair differences of adjacent samples in the deep ice. Experiments using old EDC ice that has been stored for an extended time at higher storage temperature may help to test this hypothesis in the future.

In summary, apart from analytical issues to achieve the required high precision and accuracy in atmospheric values of

elemental ratios, also fractionation/gas loss effects have to be taken into account when interpreting MOT values in deep ice cores. As observed earlier for $\delta O_2/N_2$, the gas composition in brittle ice from the BCTZ suffers from post-coring gas loss effects, while pure bubble ice provides good storage conditions for permanent gases. MOT values from pure clathrate ice below the BCTZ but well above the bedrock ($> 400$ m) show little signs of post-coring gas loss and provide reliable MOTs for all three elemental ratios. In the warm deep ice closer to the bedrock, we observed an apparent Xe enrichment and the Xe/Kr

and Xe/$N_2$ ratios did not allow us to derive reliable MOT values. The consistency of $\delta Kr/N_2$ values over the entire record and particularly their reproducibility within the analytical uncertainty gives us confidence that reliable MOT values can be derived from this parameter throughout the EDC ice core.

## 4    MOT snapshots of peak glacial and interglacial periods during the last 700 kyr

Following the discussion about gas loss above, MOTs derived from $\delta Xe/N_2$ and $\delta Xe/Kr$ show significant anomalies in samples

from the BCTZ and especially from very deep ice as shown in Figure 6.

Based on the good reproducibility of $\delta Kr/N_2$ values within the analytical uncertainty throughout the record, we conclude that the MOT derived from $\delta Kr/N_2$ is not significantly affected by the gas loss described in chapter 3 and shows consistent results throughout the record. One sample at 638 kyr BP is, however, considered an outlier based on two reasons: The sample has a





pair difference in $\delta$Kr/N$_2$ which is larger than three sigma of the reproducibility and the order of the three proxies indicates a substantial xenon loss in contrast to all other samples from pure clathrate ice. In the following, we neglect the value at 638 kyr BP and restrict our discussion of past MOT changes on the values derived from $\delta$Kr/N$_2$ as presented in Figure 9.

Despite the point-wise character of our data, the $\delta$Kr/N$_2$ derived MOT record generally supports the main findings from the
deep ocean temperature reconstruction of Elderfield et al. (2012) and Shakun et al. (2015) but in contrast to these studies our record is based on a purely physical proxy and by definition is globally fully representative for MOT. The main findings are:

(i) some episodes during the interglacial periods MIS 5.5 and MIS 9.3 are significantly warmer than the Holocene.

(ii) glacial MOT are similar throughout the last 800 kyr, although glacials prior to 450 kyr BP appear slightly colder.

(iii) the MOT during interglacial periods prior to 450 kyr BP are significantly colder than after the Mid-Brunhes event.

Note that each of our samples only represents a MOT snapshot representative of a time interval of a few centuries (Fourteau et al., 2019) given by the width of the gas age distribution for the EDC ice core. Thus, individual samples are not representative of the mean interglacial (or glacial) value, but may be subject to the millenial-scale variability of MOT. We stress that MOT is strongly influenced by changes in the Atlantic Meridional Overturning Circulation (AMOC) (Baggenstos et al., 2019; Shackleton et al., 2020) and can therefore vary on multi-centennial to millennial time scales as also seen in coupled climate
model experiments (Pedro et al., 2018; Galbraith et al., 2016). A sudden slow-down of the AMOC leads to a long-lasting accumulation of heat in the ocean interior (Pedro et al., 2018; Galbraith et al., 2016; Ritz et al., 2011), while a recovery of the AMOC slowly lowers the MOT. Accordingly, samples very close to the onset of interglacials, which may still be affected by a previously reduced AMOC, may still show elevated MOTs. The heat stored in the global ocean is only removed in the course of 2000-3000 years into the subsequent interglacial after strengthening of the AMOC. This is clearly observed in higher
resolution data for MIS5.5 (Shackleton et al., 2020), which allowed to temporally resolve this transient feature. Such a deep water mass reorganization that marks the onset of the subsequent interglacial period is also inferred from changing ocean water tracer distributions (Deaney et al., 2017).

In line with this, our MOT data based on $\delta$Kr/N$_2$ from the onset of MIS 5.5 are on average 1.0 ± 0.4 °C warmer than the Holocene and decline to interglacial values only 0.7°C warmer than the Holocene later in MIS5.5. This is in agreement with
the findings from Shackleton et al. (2020) that show significantly warmer MOTs reaching its maximum value of 1.1 ± 0.3 °C at about 129 kyr BP. Shackleton et al. (2020) attribute the early last interglacial maximum in MOT to the weakening of the AMOC that occurred over Termination II during the major iceberg discharge event from Hudson Strait in the North Atlantic commonly known as Heinrich event 11 (Capron et al., 2014). The subsequent release of heat from the ocean after AMOC resumed led to a MOT decrease of $\sim$ 1°C with stable MOT only reached after 127 kyr BP.

A similar but even more pronounced ocean warming feature is found in our new data during interglacial MIS9.3. Here the samples later in the interglacial (329 and 330 kyr BP) show within uncertainty the same temperatures as the Holocene, however, the very early interglacial samples (334 and 335 kyr BP) show 1.8 ± 0.4 °C warmer MOTs than the Holocene. In line, global deep ocean temperatures for early MIS9.3 compiled from marine sediment records in the study by Shakun et al. (2015) are intermittently nearly 2°C warmer than the Holocene. Our two early MIS9.3 samples occur also in parallel to a rapid CO$_2$ and
CH$_4$ increase at the very end of termination IV. At the onset of interglacial MIS 9.3, Nehrbass-Ahles et al. (2020) measured







**Figure 9.** MOT record relative to the Holocene from the Kr/N₂ ratios in comparison with other climate records. The EDC record is based on 85 samples, displayed as orange boxes, with the exception of one outlier at 638 kyr BP, which was printed in grey. The uncertainty is shown as described in section 2.3. a) Deep ocean reconstructions from a global compilation of 49 paired planktic $\delta^{18}O$ records are shown in darkblue (Shakun et al., 2015). Given in cyan is the benthic Mg/Ca derived temperature from Elderfield et al. (2012). b) The $CO_2$ record is based on measurements from the EDC and Vostok ice core (Nehrbass-Ahles et al., 2020; Bereiter et al., 2015; Lüthi et al., 2008; Petit et al., 1999). c) Antarctic temperatures are taken from Parrenin et al. (2013). Red shaded bars and MIS numbers on top indicate interglacials according to Masson-Delmotte et al. (2010).





the highest $CO_2$ values during the last 800 kyr, coeval with the highest $CH_4$ levels (Loulergue et al., 2008) and warm Antarctic temperatures, providing an excellent chronological constraint for the timing of AMOC resumption. An intriguing hypothesis for the high values at the onset of interglacial MIS 9.3 is therefore that the MOT overshoot observed at the final phases of terminations IV is also a result of a long period of freshwater induced AMOC suppression and deep water mass reorganization

(Ganopolski and Brovkin, 2017) that only ended with the AMOC resumption at that time. This suggests, similar to the case in MIS 5.5, that MOT was still elevated by the reduced AMOC in the millennia before and was able to regain interglacial levels several millennia after the rapid $CO_2$ and $CH_4$ increases. This emphasizes the transient character of this feature and the role of increasing AMOC on Antarctic and ocean temperature as well as on marine and terrestrial biogeochemical cycles.

Our data show glacial mean ocean temperatures very similar throughout the last 800 kyr. On average our measured glacial

MOT are 3.3 ± 0.4 °C colder than the Holocene. This is in agreement with the deep ocean anomaly derived from Elderfield et al. (2012) based on a sediment record from Chatham rise east of New Zealand. There appears to be a slight increasing trend in glacial MOT over the last 700 kyr that is also suggested by the global ocean data compilation of Shakun et al. (2015), however, the trend in our data is of the same order as the uncertainty of our reconstruction. Accordingly, we refrain from interpreting this trend at this point of time; more and higher resolution MOT data have to show whether this feature persists.

Our data also support the finding of lukewarm interglacials prior to the Mid Brunhes Event in MOT 450 kyr ago (Shakun et al., 2015; Elderfield et al., 2012) as also seen in Antarctic temperatures (EPICA community members, 2004), but also in lower interglacial concentrations of the greenhouse gases $CO_2$ (Bereiter et al., 2015; Lüthi et al., 2008) and $CH_4$ (Loulergue et al., 2008) at least for MIS13-17. Thus, a potential compensation of the excess effective energy not used for ice sheet melting during lukewarm interglacials, as discussed in the Introduction, can be ruled out based on our ice core and the marine

sediment data. Accordingly, the lukewarm interglacials must be connected to a net change in the radiative balance of the globe compared to later warmer interglacials. The MOT anomaly relative to the Holocene during the interglacials MIS 13, 15.1, 15.5 and 17 is -1.6 ± 0.4 °C, -1.7 ± 0.4 °C, -1.3 ± 0.4 °C and -1.8 ± 0.4 °C, respectively. Interglacial bottom-water temperatures increased also by about 1°C after  450 kyr BP (Elderfield et al., 2012; Shakun et al., 2015), however, Holocene bottom-water temperatures seem to be colder again than during the three preceding interglacials MIS 5.5, MIS9.3 and MIS11. Using the

global benthic $\delta^{18}O$ stack by Lisiecki and Raymo (2005), Bintanja et al. (2005) reconstructed global ice volume changes and the contributions of individual ice sheets back in time. Again, the lukewarm interglacials are significantly different compared to the later interglacials in this model reconstruction and are characterized by up to 20 m lower sea level compared to the Holocene with the largest contribution to this sea level reduction caused by larger ice sheets in Eurasia. Note, however, that sea levels lower than in the Holocene during lukewarm interglacials can be observed in some but not all sea level reconstructions from

marine sediments (Rohling et al., 2014; Spratt and Lisiecki, 2016). In any case, an increase in ice volume during lukewarm interglacials cannot be easily attributed to changes in orbital forcing, as the effective energy at high northern latitudes (Tzedakis et al., 2017) during these intervals did not significantly differ from the insolation conditions during later interglacials. Thus, the lukewarm interglacials require additional changes in the Earth System that prevented northern hemisphere ice sheets from retreating and temperatures to increase to the same extent as in later interglacials.





Irrespective of the ultimate cause for the incomplete deglaciation during the lukewarm interglacials, our significantly lower MOT values during these lukewarm interglacials provide robust evidence for a significantly altered energy budget of our planet compared to the Holocene. Note that a MOT lowering of about 1°C translates into a decrease in the ocean heat content of 5-6· $10^{24}$ J, or about 30% of our typical glacial/interglacial change in heat content. We can estimate whether the deviation in

climate boundary conditions (greenhouse gas concentrations and albedo) are sufficient to explain the heat deficit. Lukewarm interglacial $CO_2$ concentrations are typically 245 ppm (instead of 280 ppm in later interglacials) and $CH_4$ concentrations 620 ppb (instead of 700 ppb thereafter). This translates into a radiative forcing of about -0.79 W/m² (Myhre et al., 1998) and using a recent paleo-based value of the equilibrium climate sensitivity of 3.8°C for a doubling of $CO_2$ (Tierney et al., 2020) (which includes fast feedbacks such as sea ice coverage) into a cooling of the global mean surface temperature (GMST) of about

0.8°C. In addition, we also need to take into account the long term Earth System feedbacks such as planetary albedo increase due to extended land ice coverage. For the LGM, the fraction of greenhouse gas forcing to total forcing has been estimated to approximately 0.4 (Baggenstos et al., 2019). Using the greenhouse gas forcing for the lukewarm interglacials as given above, the estimated total forcing is thus close to 2 W/m², which yields an expected GMST decrease of 2.0°C for the lukewarm interglacials. However, comparing global surface temperature to MOT is not straighforward (Bereiter et al., 2018b), because

(i) MOT is biased towards sea surface temperatures in deep water formation regions, for example in the North Atlantic, and thus subject to polar amplification. And (ii) because land temperature change is poorly known and cannot easily be linked to MOT change. However, a recent study has tightly constrained LGM GMST change to 6°C (Tierney et al., 2020), which goes along with approximately 3°C MOT change for the LGM. Assuming this relationship holds also for earlier interglacials, we can scale our average MOT decrease of 1.6°C into a GMST reduction of 3.2°C. This is somewhat larger than the expected

GMST lowering of 2.0°C due to the radiative forcing, and suggests that either the ratio of greenhouse gas forcing to total forcing or the scaling from MOT to GMST were different during the lukewarm interglacials. Note, however, that both of these factors are poorly constrained, and e.g. using 0.3 instead of 0.4 for the forcing ratio along with a slightly smaller MOT/GMST scaling would bring the two estimates into agreement. Thus, our MOT lowering during lukewarm interglacials is within the range expected from the radiative forcing caused by the lowered $CO_2$ and $CH_4$ in those intervals and the connected longer-term

Earth System feedbacks.

The close relationship of MOT and global radiative forcing by changing greenhouse gas concentrations is not only valid for lukewarm vs. full interglacials but to first order holds for the entire noble gas based MOT record available so far. Bereiter et al. (2018b) showed that MOT is closely correlated with Antarctic temperature and the MOT evolution is essentially synchronous with $CO_2$ during the last transition. Our record from the last 40 kyr supports this correlation. The scatter plot presented in Figure

10 shows the correlation between MOT and logarithmic $CO_2$, as expected from the radiative forcing of surface temperatures by $CO_2$ concentration (IPCC, 2013). The squared correlation coefficient for all the reconstructed MOTs from EDC and logarithmic $CO_2$ suggests that 74% of the variance in the MOT record can be explained by changing $CO_2$ and the accompanying Earth System feedbacks. Taking a closer look at Figure 10 we see again that the peak glacials and the lukewarm interglacials prior to 450 kyr BP show approximately 1°C colder temperatures than expected from the $CO_2$ relationship derived for the last 40



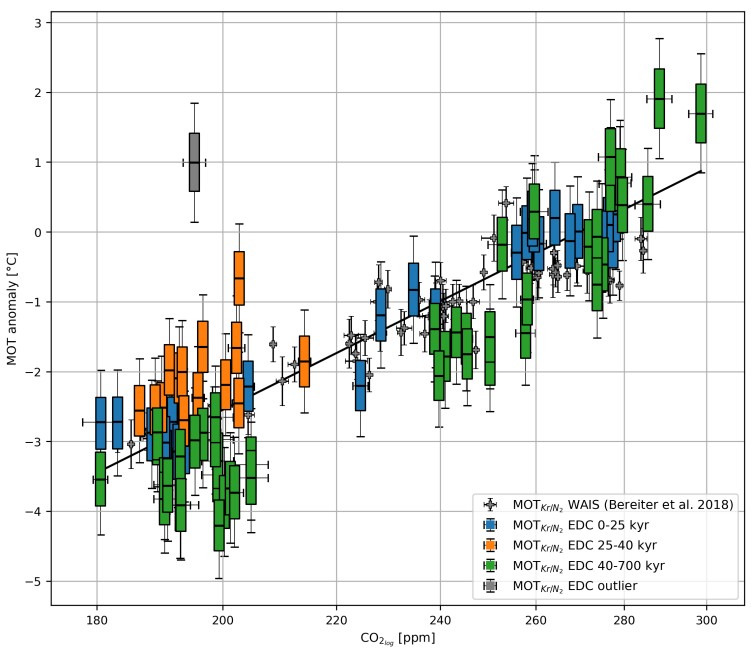

**Figure 10.** Scatter plot showing a linear relationship between logarithmic $CO_2$ (values taken from Bereiter et al. (2015)) and the MOT relative to the Holocene. MOT reconstructions are based on $Kr/N_2$ and corrected for thermal, gravitational and HID fractionation. MOTs from the last 25 kyr are shown in blue, samples from 25 - 40 kyr BP are shown in orange and in green samples from 120 to 700 kyr ago. The linear regression (black) is based on EDC data only, and has an $r^2$ of 0.74. Given in grey is the MOT reconstruction over the last 22 kyr from Bereiter et al. (2018b) relative to the Holocene.

kyr. In view of the considerable systematic uncertainties introduced by the correction of the data, we refrain however from speculating on a possible cause for these lower temperatures at this point.

## 5 Conclusions

Our study presents MOT snapshots of peak glacial and interglacial periods during the last 700 kyr. The comparison of MOT
5 values derived from different elemental ratios revealed systematically depleted MOTs based on $\delta Xe/N_2$ and $\delta Xe/Kr$ values from the BCTZ and enriched MOTs from $\delta Xe/N_2$ and $\delta Xe/Kr$ in the ice deeper than 2500 m. We attribute this to a Xe depletion during clathrate formation through its low permeation coefficient and/or reduced incorporation of Xe in sII clathrates in the ice. This leads to an enrichment of Xe in bubbles in the BCTZ and a depletion of Xe in reforming PLIs in the deep ice. Together with a gas loss through postcoring microcracks, this leads to a depletion of Xe relative to $N_2$ and Kr in the BCTZ and an
10 enrichment in the deep ice. In addition, continuing work on understanding the firn gas and especially the HID are crucial to reduce the uncertainties of the MOT reconstructions based on noble gases.





Despite these limitations we find that glacial MOTs based on $\delta$Kr/N$_2$, which appear not significantly influenced by gas loss/fractionation processes, are on average 3.3 ± 0.4 °C colder than the Holocene throughout this noble gas record and that the MOT during interglacial periods prior to 450 kyr BP are on average 1.6 ± 0.4 °C lower compared to the Holocene. Episodes during the onsets of interglacials MIS 5.5 and 9.3 are significantly warmer than the Holocene and can be attributed to the ocean

heat uptake during preceding times of lowered AMOC. The most pronounced feature of the record is the significantly increased MOT at the onset of MIS 9.3, which is coeval with the CO$_2$ and CH$_4$ overshoot at that time and reflects maximum ocean heat uptake before the AMOC resumption.

Clearly, our data provide only snapshots of MOT in glacial and interglacial conditions, which, however, define the natural range in which the global heat budget has changed over the last 700 kyr. Our data show that the range in MOT can be quantita-

tively explained by changes in global radiative forcing (due to changes in greenhouse gas concentrations) taking into account long-term Earth System feedbacks and that MOT is biased towards sea surface temperature changes in high latitudes. More high-resolution and high-precision MOT data from noble gases in ice cores (avoiding gas loss issues) and a better understanding of the firn fractionation processes affecting the gas composition in ice cores are needed to make use of the full potential of noble gas derived MOT in future studies.

*Code and data availability.* Noble gas data derived in this study will be made available on the NOAA paleoclimate database and on PANGAEA. The Python code used for the correction of the data will be made available on request.

*Author contributions.* The concept of this study was created by HF, JS and TK, the research was funded through third party funding by HF. The analytical method was established by TK, DB and MH with input by JS and HF. The sample analyses were performed by MH and DB. Data evaluation was performed by MG and MH with inputs from DB and HF. The manuscript was written by HF and MH, with support by

all coauthors, who also contributed heavily to the discussion of the data.

*Competing interests.* The authors declare that they have no conflict of interest.

*Acknowledgements.* We thank Eric Wolff, Gregory Teste and Catherine Ritz for their help with sample selection and cutting. C. Ritz kindly provided borehole temperature profiles at Dome C. Most helpful discussions on noble gases as well as gas and heat transport in the firn with Jeff Severinghaus, Benjamin Birner, Sarah Shackleton, Bernhard Bereiter, and Jakob Schwander are gratefully acknowledged. The

research leading to these results has received funding from the European Research Council (ERC) under the European Union's Seventh Framework Programme FP7/2007-2013 ERC Grant 226172 [ERC Advanced Grant Modern Approaches to Temperature Reconstructions in Polar Ice Cores (MATRICs)] and the Swiss National Science Foundation (grant numbers 200021-155906, 200020-172506). This work is a contribution to EPICA, a joint European Science Foundation/European Commission scientific program funded by the European Union and



national contributions from Belgium, Denmark, France, Germany, Italy, The Netherlands, Norway, Sweden, Switzerland, and the United Kingdom. The main logistic support was provided by Institut Polaire Français Paul-Emile Victor (IPEV) and PNRA (at Dome C) and AWI (at EDML). This work is EPICA publication no. xxx



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
