# Peer review of "Snapshots of mean ocean temperature over the last 700,000 yr using noble gases in the EPICA Dome C ice core"

_Climate of the Past, 2020_

## Referee Comment (RC1) · Anonymous Referee #1 · 27 Nov 2020

The reconstruction of mean ocean temperature from past gas composition from ice cores is very complicated and tricky. Numerous corrections need to be applied and the authors go through great length to explain what they do and why. I understand that they want to be maximum transparent on the method they use. However, the manuscript is very long, and requires endurance to read. It would profit from being split into a main text and an appendix section with all the technical details. A sketch in isotope space showing the various corrections and their magnitude along with the respective effect on MOT would be useful. The temperature gradient in the firn layer is an important correction. The authors favor a model based approach for that correction that fits the long term average of the individual reconstructions based on the data. This

I find troublesome. From the denser measurements up to 40 kyr BP it looks like the signal is not random.

Specific comments: Page 4 line 17.. : How is Kr affected by drill fluid when all other components have been gettered away? Page 8, lines 11-17: Instead of writing DT is negative write that the temperature is higher at depth due to geothermal heat flow (or do I misunderstand what is said here?) Figure 3: Please lower the top tags slightly so they do not interfere with the frame. Page 14, last paragraph: First, you argue that there may be a signal in the data then you invalidate that statement but do not say it. Page 18, line 7,8: What is the argument to assume no change in the saturation state?

---

## Referee Comment (RC2) · Jeff Severinghaus (Referee) · 25 Dec 2020

This manuscript describes a heroic effort to use noble gases from the full 700-kyr EPICA Dome C ice core to infer past mean ocean temperature, based on the well-known temperature dependence of noble gas solubility in the ocean. The method takes advantage of the fortunate fact that the total amount of N2, Kr, and Xe in the combined ocean-atmosphere system is remarkably stable over million-year timescales, at a sufficiently high level that they can be assumed to be unchanging.

The difficulty that had to be overcome by the authors is substantial. Many unforseen artifacts, such as gas loss and clathrate based issues, had to be wrestled with. This

work truly pioneered the use of noble gases in very deep ice cores where geothermal heat made the ice core rather warm, and depressurization effects upon core recovery were extreme. Transport issues further vexed the effort, including failures of the cooling system that allowed the ice cores to get warm. Fractionation mechanisms are still incompletely understood in ice cores, leading to small disagreements between the three gas pairs used.

Nonetheless, the authors persevered and the result is a spectacular advance in scientific understanding of the behavior of the planetary energy imbalance and ocean dynamics over the late Pleistocene ice ages. This is truly an excellent piece of science and a carefully and thoroughly executed and painstaking research tour de force. It goes without saying, then, that this manuscript should be published with only very minor revisions.

I have attached a copy of the manuscript with my suggested edits in red. One area that needs a re-write is the paragraph on air clathrates, which seems to have been influenced by prior work done on Greenland ice. Antarctic ice has lower impurity content (and thus clathrate nucleation sites) than Greenland ice, and therefore has clathrates that are fewer in number than the number of bubbles, requiring air to permeate some distance through the ice lattice from the air bubble to the (relatively rare) growing clathrate. This nucleation limitation effect is not seen in Greenland ice to my knowledge.

To the authors: well done! This is a beautiful piece of science and will no doubt have lasting value.

Please also note the supplement to this comment:
https://cp.copernicus.org/preprints/cp-2020-127/cp-2020-127-RC2-supplement.pdf

**Supplement:**

[Figure]

comments in red

[revised manuscript text omitted]

*Shackleton et al. 2019 documented artifacts in the lower end of the WAIS BCTZ analogous to the "Luthi effect" seen in CO2.*
*NGRIP has high impurity content so is not a good analogue for EDC.*

In the following we will try to  explore the processes that we think are responsible for this fractionation. Studies of air bubbles and clathrate inclusions on the NGRIP ice core reveal two stages of clathrate formation: primary bubble clathratization and secondary clathrate growth (Kipfstuhl et al., 2001). Primary  air clathratization takes place in the upper part of the

*!nucleation limitation prevents most individual bubbles from clathrating in east Antarctica!*

BCTZ, where primary clathrates are formed  from  air in nearby bubbles. The partial pressures of the individual gases in the bubbles increase linearly with depth due to the increasing hydrostatic pressure. In contrast, the partial pressures of the gases forming the primary clathrates is controlled by the dissociation pressure of the air clathrate,

*this seems slightly incorrect - "air" acts as a solid solution of O2 and N2*

*? this doesn't seem right - early forming hydrates have O2 in them.*

[revised manuscript text omitted]

---

## Author Comment (AC1) · 22 Jan 2021

Dear referees, dear editor

Below you'll find our final response to the reviewer comments. We are most grateful for the positive evaluation of our work by both referees. We are especially thankful for the euphoric assessment by Jeff Severinghaus. This means a lot to us, as Jeff is the pioneer of noble gas thermometry on ice cores.

We outline how we plan to meet the reviewer comments and what textual changes we plan in the revised version of our manuscript. We will not go through any of the grammar or typo corrections at this point but will carefully revise our manuscript also in this respect based on the minor comments by the referees. Here we concentrate on textual or argumentation changes. Our answer is split according to referee #1 and #2 where the review comments are given in red and our reply in black.

**Referee #1**

The reconstruction of mean ocean temperature from past gas composition from ice cores is very complicated and tricky. Numerous corrections need to be applied and the authors go through great length to explain what they do and why. I understand that they want to be maximum transparent on the method they use. However, the manuscript is very long and requires endurance to read. It would profit from being split into a main text and an appendix section with all the technical details.

We are well aware of the level of detail we provide in this manuscript and that for those readers only interested in the final results it may not so easy to digest. However, we regard this paper as a reference document also for future studies on MOT using noble gases in ice cores on the EDC and other ice cores, as a similar reference document does not exist yet in a peer-review publication, which provides all the detail that is needed to replicate the results. Accordingly, we would like to keep the discussion of noble gas corrections and gas loss issues in the manuscript. As we do not have a purely mathematical derivation in the manuscript either, it appears also difficult to us to transfer some of the material into an appendix. However, to make navigating this manuscript easier for all readers (whether they are interested in methodological aspects, air hydrates or just in the MOT numbers), we will add a paragraph at the end of the Introduction that explains the structure of the paper and refers to the sections that are of interest for a specific reader.

A sketch in isotope space showing the various corrections and their magnitude along with the respective effect on MOT would be useful.

We will add some information on the effect of the various corrections on MOT in Fig. 2

The temperature gradient in the firn layer is an important correction. The authors favor a model based approach for that correction that fits the long term average of the individual reconstructions based on the data. This I find troublesome. From the denser measurements up to 40 kyr BP it looks like the signal is not random.

Fig. 4 displays the firn temperature gradient that we derive either using the model (model-based approach) or using the isotopic values only (data-based approach). In the model case (red squares) we see a small change in the firn temperature as expected, as the surface temperature at EDC was 8-10 °C colder in the LGM than in the Holocene while the

temperature at the bedrock remained at the pressure melting point. Accordingly, the overall temperature difference between surface and bedrock increases in the glacial and hence also the temperature difference between surface and close-off depth. **Note that in the model-based approach we include this systematic variation in ΔT in our correction,** so referee #1 does not have to worry for this model-based approach.

In the data-based approach one may see also some systematic variation in ΔT (black squares in Fig. 4), however (i) the firn temperature difference and its variations are unphysically large nor can (ii) positive temperature differences physically occur at Dome C. The variation seen in the data points over the last 40 kyr is of the same size as the analytical error thus should not be interpreted. In the manuscript we only used the mean of the data to get a representative mean kinetic fractionation using our data-based approach to check the consistency of the results of the two approaches. Using this mean ΔT leads to a mean kinetic correction that is in line with the model-based approach. However, even using the mean ΔT our data-based MOT reconstruction is subject to too much analytical error to allow meaningful conclusions in terms of MOT changes. Hence, in the end we discarded the data-based approach for MOT reconstruction and the use of a mean ΔT in the data-based approach, criticized by the referee, is not included in any of the final results or the conclusions of the paper. We will revise the text accordingly, to stress these points.

Specific comments: Page 4 line 17. : How is Kr affected by drill fluid when all other components have been gettered away?

When we first saw our results, we shared the astonishment of referee #1 that any contamination may survive the gettering process, however, the data clearly shows that samples showing anomalies in d15N in the ungettered aliquot show also anomalies in 82Kr in the gettered aliquot. We have no conclusive evidence yet what is causing this interference, however, we are currently working on lab experiments to get more insight on this. Either the zoo of higher organic compounds in the drill fluids allows for some component to be not completely gettered if the drill fluid contamination is too large or the H2 released by the gettering of such organic compounds (and which may not be quantitatively trapped by getter material if its abundance is too high) leads to chemical effects in the source that cause the mass 82 interference. We will elaborate a little bit more on this in the revised version but cannot provide an ultimate answer at this point.

Page 8, lines 11-17: Instead of writing DT is negative write that the temperature is higher at depth due to geothermal heat flow (or do I misunderstand what is said here?)

will do

Figure 3: Please lower the top tags slightly so they do not interfere with the frame.

will do

Page 14, last paragraph: First, you argue that there may be a signal in the data then you invalidate that statement but do not say it.

as outlined above we will discuss this in more detail to justify our approach

Page 18, line 7,8: What is the argument to assume no change in the saturation state?

We refrain from including a change in the saturation state as no experimental evidence for the saturation during glacial times exist. In fact, Bereiter et al. (2018) argued for an increase in saturation due to reduced ocean overturn, however, the increased sea ice coverage especially in the Southern Ocean could also argue for a decrease in saturation. Thus, no robust assumption can be made on the change in saturation state. We will add a paragraph on this in the final manuscript.

**Referee #2**

This manuscript describes a heroic effort to use noble gases from the full 700-kyr EPICA Dome C ice core to infer past mean ocean temperature, based on the well-known temperature dependence of noble gas solubility in the ocean. The method takes advantage of the fortunate fact that the total amount of N2, Kr, and Xe in the combined ocean-atmosphere system is remarkably stable over million-year timescales, at a sufficiently high level that they can be assumed to be unchanging. The difficulty that had to be overcome by the authors is substantial. Many unforeseen artifacts, such as gas loss and clathrate based issues, had to be wrestled with. This work truly pioneered the use of noble gases in very deep ice cores where geothermal heat made the ice core rather warm, and depressurization effects upon core recovery were extreme. Transport issues further vexed the effort, including failures of the cooling system that allowed the ice cores to get warm. Fractionation mechanisms are still incompletely understood in ice cores, leading to small disagreements between the three gas pairs used. Nonetheless, the authors persevered and the result is a spectacular advance in scientific understanding of the behavior of the planetary energy imbalance and ocean dynamics over the late Pleistocene ice ages. This is truly an excellent piece of science and a carefully and thoroughly executed and painstaking research tour de force. It goes without saying, then, that this manuscript should be published with only very minor revisions.

we are very grateful for this positive evaluation of our work

I have attached a copy of the manuscript with my suggested edits in red. One area that needs a re-write is the paragraph on air clathrates, which seems to have been influenced by prior work done on Greenland ice. Antarctic ice has lower impurity content (and thus clathrate nucleation sites) than Greenland ice, and therefore has clathrates that are fewer in number than the number of bubbles, requiring air to permeate some distance through the ice lattice from the air bubble to the (relatively rare) growing clathrate. This nucleation limitation effect is not seen in Greenland ice to my knowledge.

We agree with the referee that mixing the observations on clathrate formation made in Greenland (e.g. Kipfstuhl) and in Antarctica (e.g. Uchida) was a bit confusing. Accordingly, we will rewrite and extend this discussion to base it entirely on the work of Uchida et al., 2011 (and references therein), who use samples from Dome Fuji (which are very similar in terms of climate boundary condition as those from Dome C). These results clearly show that in the BCTZ of Dome Fuji clathrate nucleation is slow and that early nucleating air hydrates grow by permeation of air from coexisting bubbles, while at the same time the number of hydrates increases due to successive nucleation of new hydrates. In the deep, fully clathrated

ice, hydrates grow as well, but their number is declining. Here their total number is declining by an Ostwald ripening process, where air permeates from smaller hydrates to larger ones.

To the authors: well done! This is a beautiful piece of science and will no doubt have lasting value.
Please also note the supplement to this comment:
https://cp.copernicus.org/preprints/cp-2020-127/cp-2020-127-RC2-supplement.pdf

Here we will not list grammar or typo corrections suggested by referee #2 but will correct them in the revised version. We will shortly respond to main points made in the annotated manuscript

equation (1): we will clarify the units

page 8: we will recalculate the values using the local gravitational acceleration

page 9 decrease/increase issue. We apologize that our wording is unclear. We agree with referee #2 but to avoid any confusion, we will delete this sentence

page 13: we will include the comment of the referee about a potential sampling artefact at the WAIS firn pumping as (Jeff Severinghaus, personal communication)

page 20: we will revise the discussion on clathrate formation and growth and the accompanying permeation processes as outlined above

---

## Author Response (AR1)

Dear referees, dear editor

Below you'll find our point-to-point response to the reviewer comments. We are most grateful for the positive evaluation of our work by both referees. We are especially thankful for the euphoric assessment by Jeff Severinghaus. This means a lot to us, as Jeff is the pioneer of noble gas thermometry on ice cores.

Provided below you find a point-to-point reply to reviewer comments. In addition, reviewer Severinghaus provided an annotated pdf-file with some language improvements. We will not list these grammar or typo corrections at this point but we have carefully revised our manuscript also in this respect based on the minor comments by the referees (see also the differential Latex file, showing the differences of the revised and initially submitted manuscript). Here we concentrate on textual or argumentation changes. Our answer is split according to referee #1 and #2, where the review comments are given in red, our reply in black and new text is indicated in blue.

**Referee #1**

The reconstruction of mean ocean temperature from past gas composition from ice cores is very complicated and tricky. Numerous corrections need to be applied and the authors go through great length to explain what they do and why. I understand that they want to be maximum transparent on the method they use. However, the manuscript is very long and requires endurance to read. It would profit from being split into a main text and an appendix section with all the technical details.

We are well aware of the level of detail we provide in this manuscript and that for those readers only interested in the final results it may be not so easy to digest. However, we regard this paper as a reference document also for future studies on MOT using noble gases in ice cores on the EDC and other ice cores, as a similar reference document does not exist yet in a peer-review publication, which provides all the detail that is needed to replicate the results. In particular, we explain for the first time the kinetic fractionation correction in detail, which adds another layer of complexity.

Accordingly, we would like to keep the discussion of noble gas corrections and gas loss issues in the manuscript. As we do not have a purely mathematical derivation in the manuscript either, it appears also difficult to us to transfer some of the material into an appendix.

However, to make navigating this manuscript easier for all readers (whether they are interested in methodological aspects, air hydrates or just in the MOT numbers), we added a paragraph at the end of the Introduction that explains the structure of the paper and refers to the sections that are of interest for a specific reader:

The manuscript is organized as follows: In chapter 2 we describe the overall analytical procedure and uncertainty to obtain noble gas isotopic and elemental ratios and the corrections that have to be applied to correct for systematic transport effects in the firn column. As EDC is a very low accumulation site, which exhibits a permanent firn temperature gradient larger than 1°C, thermal diffusion has to be precisely quantified. Moreover, we describe for the first time in detail the correction of kinetic fractionation by non-diffusive transport adding an additional layer of complexity and uncertainty. Another process acting on the noble gas composition in the EDC ice core is post-coring gas loss that affects samples from the Bubble Clathrate Transition Zone and, more severely, fully clathrated ice from the

deepest ice at Dome C which is close to the pressure melting point. These effects are for the first time described in detail in chapter 3 and a hypothesis how these lead to systematic noble gas fractionation is presented. For the reader only interested in the final results of our MOT reconstruction over the last 700 kyr and their discussion we refer directly to chapter 4 and the Conclusions.

A sketch in isotope space showing the various corrections and their magnitude along with the respective effect on MOT would be useful.

We added the information on the range of the effect that each individual correction has on the final absolute MOT values to Fig. 2. We extended the figure caption by:

Here, we correct for the influences of sea level (SL) change and the saturation state (SAT) of the global ocean. In a last step, the resulting MOT are corrected for their Holocene offset (see text for details). The approximate range of the respective correction performed on all our samples on MOT values can be found in the colored boxes. Positive values (red boxes) imply that the correction leads to an increase in absolute MOT values. It is important to stress that the size of the correction is not a measure of the uncertainty associated to it. The gravitational enrichment correction, for example, dominates all other corrections by orders of magnitude. Nonetheless, the analytical uncertainty of that correction is only minor, as it is largely given by the uncertainty in diffusive column height, the exact determination of which is one of our analytical foci. Also note that it is the size of the range of a correction, rather than the absolute value of the correction, that has an influence on glacial-interglacial MOT differences. For example, although the correction of elemental ratios for kinetic fractionation appears to be the second-most influential correction, when considering its absolute value, its range is only narrow. Thus, it alters glacial-interglacial MOT difference only little.

The temperature gradient in the firn layer is an important correction. The authors favor a model based approach for that correction that fits the long term average of the individual reconstructions based on the data. This I find troublesome. From the denser measurements up to 40 kyr BP it looks like the
signal is not random.

Fig. 4 displays the firn temperature gradient that we derive either using the model (model-based approach) or using the isotopic values only (data-based approach). In the model case (red squares) we see a small change in the firn temperature as expected, as the surface temperature at EDC was 8-10 °C colder in the LGM than in the Holocene while the temperature at the bedrock remained at the pressure melting point. Accordingly, the overall temperature difference between surface and bedrock increases in the glacial and hence also the temperature difference between surface and close-off depth. **Note that in the model-based approach we include this systematic variation in ΔT in our correction for each individual sample**.

In the data-based approach one may see also some systematic variation in ΔT (black squares in Fig. 4), however (i) the firn temperature difference and its variations are unphysically large nor can (ii) positive temperature differences physically occur at Dome C. The variation seen in the data points over the last 40 kyr is of the same size as the analytical error thus should not be interpreted. In the manuscript we only used the mean of the data to get a representative mean kinetic fractionation using our data-based approach to check the consistency of the results of the two approaches. Using this mean ΔT leads to a mean kinetic correction that is in line with the model-based approach. However, even using the mean ΔT our data-based MOT

reconstruction is subject to too much analytical error to allow meaningful conclusions in terms of MOT changes. Hence, in the end we did not use MOT data based on the data-based approach for our interpretation of past ocean temperature changes. Accordingly, the use of a mean ΔT in the data-based approach, criticized by the referee, is not included in any of the final results or the conclusions of the paper. In section 2.2.2. b) we added on page 15:

The very high scatter in data-derived ΔT is displayed in Fig. 4. The standard deviation of ΔT over the last 40 kyr is 6°C, which is of the same order as the expected analytical uncertainty in ΔT of 8°C derived from error propagation. Hence, this large scatter can be explained by analytical noise and shows that any systematic variation in ΔT over the last 40 kyr cannot be reliably quantified using the data-based approach.

and at the end of the section we added:

In summary, the data-based approach shows that we can quantitatively correct the gas transport related fractionations that occur in the firn column, however, the precision of the data-based approach is not sufficient for single samples. Accordingly, in the discussion of our final MOT data in chapter 4 we do not use the data-based approach and rely on the much more precise model-based correction.

Similar to the referee point on systematic changes in ΔT we also clarified the text on potential systematic changes in the kinetic fraction. On page 9/10 we write:

It is worth noting that the ratios of the mean kinetic fractionations over the last 40 kyr as displayed in Figure 3 agree well within uncertainties with the ratios predicted by Birner et al. (2018). Note that the kinetic fractionations for the last 40 kyr displayed in Figure 3 suggest, apart from substantial analytical scatter, also some systematic variations. As these systematic variations are generally in line with the fractionation ratios predicted by Birner et al. (2018), the changes are likely caused by changing kinetic fractionation conditions with time (stronger barometric pumping, higher wind speeds at the surface, etc.). However, some samples older than 40 kyr show much higher deviations from the mean that cannot be explained by analytical uncertainty or changing kinetic fractionation. Accordingly, we do not use the individual HID values of each sample as displayed in Figure 3b for samples older than 40 kyr to quantify $\varepsilon^*_{Ar-N2}$. Instead, we use the mean isotopic fractionation derived for the last 40 kyr and assume that this mean value is also representative for the correction of our samples older than 40 kyr. As the choice of $\varepsilon^*_{Ar-N2}$ has a systematic effect on the final reconstructed MOT, we also take the variation of $\varepsilon^*_{Ar-N2}$ over the last 40 kyr into account to quantify the systematic uncertainty introduced by this choice. Thus we use the mean kinetic fractionation over the last 40 kyr plus (minus) its standard deviation as upper (lower) bounds of the kinetic fraction to calculate MOT. We regard this as a systematic uncertainty in our MOT reconstruction and separate this uncertainty from the stochastic uncertainty introduced by the analytical error.

and on page 13 we clarify

The kinetic fractionation $\varepsilon^*_{Ar-N2}$ calculated using modeled ΔT is on average -0.009+-0.001 ‰ (1σ) over the last 40 kyr (note that the HID values given in Figure 3b represent a measure of $\varepsilon^*_{Ar-N2}$ scaled by a factor of 6.3, 4.25 and 2.05 for δXe/Ar, δKr/Ar and δXe/Kr , respectively, as derived from Birner et al., (2018)). As described above, we use the mean plus/minus the standard deviation of the kinetic fractionation over the last 40 kyr to correct also the samples older than 40 kyr, and to assess the uncertainty introduced by this correction.

Specific comments: Page 4 line 17. : How is Kr affected by drill fluid when all other components have been gettered away?

When we first saw our results, we shared the astonishment of referee #1 that any contamination may survive the gettering process, however, the data clearly shows that samples showing anomalies in $\delta^{15}N$ in the ungettered aliquot show also anomalies in $^{82}Kr$ in the gettered aliquot. We have no conclusive evidence yet what is causing this interference, however, we are currently working on lab experiments to get more insight on this. Either the zoo of higher organic compounds in the drill fluids allows for some component to be not completely gettered, if the drill fluid contamination is too large, or the $H_2$ released by the gettering of such organic compounds (and which may not be quantitatively trapped by getter material if its abundance is too high) leads to chemical effects in the source that cause the mass 82 interference. We added the following paragraph:

Unexpectedly, we see also outliers in $\delta^{86/82}Kr$ and $\delta^{84/82}Kr$ values in the gettered larger aliquot in the same samples where we find the $\delta^{15}N$ and $\delta^{40/36}Ar$ outliers, which points to an interference at mass 82. We have no conclusive evidence yet what is causing this interference. Either the large variety of higher organic compounds in the drill fluid allows for some component to be not completely gettered, if the drill fluid contamination is too large, or the $H_2$ released by the gettering of such organic compounds (and which may not be quantitatively trapped by the getter material if its abundance is too high) leads to chemical effects in the source that causes the mass 82 interference. Further lab experiments are required to elucidate this issue. In summary, we only use $\delta^{86/84}Kr$ values in our further data evaluation.

Page 8, lines 11-17: Instead of writing DT is negative write that the temperature is higher at depth due to geothermal heat flow (or do I misunderstand what is said here?)

we changed the text do:

Here $\Delta T$ is defined as the temperature difference between the top and the bottom of the DCH. As EPICA Dome C is a low accumulation site, the mean annual firn temperature increases with depth (in the absence of temporal climate changes at the surface (Ritz et al., 1982)) due to the geothermal heat flux at the bottom and gas enclosed in bubbles is expected to be slightly depleted by thermal diffusion relative to its gravitational value.

Figure 3: Please lower the top tags slightly so they do not interfere with the frame.

done

Page 14, last paragraph: First, you argue that there may be a signal in the data then you invalidate that statement but do not say it.

as outlined above, we changed our text to clarify this

Page 18, line 7,8: What is the argument to assume no change in the saturation state?

We refrain from including a change in the saturation state as no experimental evidence for the saturation during glacial times exist. In fact, Bereiter et al. (2018) argued for an increase in saturation due to reduced ocean overturn, however, the increased sea ice coverage especially

in the Southern Ocean could also argue for a decrease in saturation. Thus, no robust assumption can be made on the change in saturation state. However, for consistency, we now included a (constant) undersaturation of Xe and Kr as estimated from recent observations. This had, however, only a marginal effect on the MOT reconstruction. We added the following text:

To reconstruct MOT values from the corrected elemental ratios, we use the ocean-atmosphere box model described by Bereiter et al. (2018b), except for one slight update, as we use the solubility equations given by Jenkins et al. (2019). The basic assumption in the model is that $N_2$, Kr and Xe are conserved in the ocean-atmosphere system and that their distribution between the two reservoirs is controlled by the temperature dependent dissolution in the ocean water. The model accounts for the effects of changes in ocean salinity, volume, and atmospheric pressure on the oceanic inventories of each of the three gases, which have individual temperature-dependent solubilities. The records of Lambeck et al. (2014) and Spratt and Lisiecki (2016) are used to reconstruct the sea level for the last 27 kyr and the time interval 28-800 kyr BP, respectively. We assume a constant undersaturation of the heavy noble gases in the ocean by 3% for Xe and 1.5% for Kr (Hamme and Severinghaus, 2007; Loose et al., 2016; Hamme et al., 2019). In contrast to Bereiter et al. (2018b) we do not impose a temporal change of this undersaturation, as we have no observational evidence about such a change in saturation state and we cannot even provide a convincing argument whether the saturation may have increased or decreased during glacial times. The overall slower ocean overturn in the glacial may suggest an increase in saturation while the expansion of sea ice (especially in the Southern Ocean) would speak for a stronger undersaturation of the heavy noble gases. Accordingly, we refrain from changing the saturation state in our model. Sensitivity analyses show that a reduction of the glacial undersaturation to half its current value will lead to a warming of our glacial MOT reconstruction by a few tens of a degree.

**Referee #2**

This manuscript describes a heroic effort to use noble gases from the full 700-kyr EPICA Dome C ice core to infer past mean ocean temperature, based on the well-known temperature dependence of noble gas solubility in the ocean. The method takes advantage of the fortunate fact that the total amount of N2, Kr, and Xe in the combined ocean-atmosphere system is remarkably stable over million-year timescales, at a sufficiently high level that they can be assumed to be unchanging.

The difficulty that had to be overcome by the authors is substantial. Many unforeseen artifacts, such as gas loss and clathrate based issues, had to be wrestled with. This work truly pioneered the use of noble gases in very deep ice cores where geothermal heat made the ice core rather warm, and depressurization effects upon core recovery were extreme. Transport issues further vexed the effort, including failures of the cooling system that allowed the ice cores to get warm. Fractionation mechanisms are still incompletely understood in ice cores, leading to small disagreements between the three gas pairs used.

Nonetheless, the authors persevered and the result is a spectacular advance in scientific understanding of the behavior of the planetary energy imbalance and ocean dynamics over the late Pleistocene ice ages. This is truly an excellent piece of science and a carefully and thoroughly executed and painstaking research tour de force. It goes without saying, then, that this manuscript should be published with only very minor revisions.

we are very grateful for this positive evaluation of our work

I have attached a copy of the manuscript with my suggested edits in red. One area that needs a re-write is the paragraph on air clathrates, which seems to have been influenced by prior work done on Greenland ice. Antarctic ice has lower impurity content (and thus clathrate nucleation sites) than Greenland ice, and therefore has clathrates that are fewer in number than the number of bubbles, requiring air to permeate some distance through the ice lattice from the air bubble to the (relatively rare) growing clathrate. This nucleation limitation effect is not seen in Greenland ice to my knowledge.

We agree with the referee that mixing the observations on clathrate formation made in Greenland (e.g. Kipfstuhl) and in Antarctica (e.g. Uchida) was a bit confusing. Accordingly, we rewrote and extended this discussion to base it entirely on the work by Uchida et al., 2011 (and references therein), who use samples from Dome Fuji (which are very similar in terms of climate boundary condition as those from Dome C). These results clearly show that in the BCTZ of Dome Fuji clathrate nucleation is slow and that early nucleating air hydrates grow by permeation of air from coexisting bubbles, while at the same time the number of hydrates increases due to successive nucleation of new hydrates. In the deep, fully clathrated ice, hydrates grow as well, but their number is declining. Here their total number is declining by an Ostwald ripening process, where air permeates from smaller hydrates to larger ones. The revised text reads as follows:

In the following discussion, we will try to motivate the processes that we think are responsible for this fractionation. Studies of air bubbles and clathrate inclusions on the Dome Fuji ice core reveal two stages of clathrate formation and clathrate growth in cold Antarctic ice (Uchida et al., 2011). Clathrate formation starts when the bubble pressure, which increases linearly with depth due to the increasing hydrostatic pressure, reaches the dissociation pressure of air hydrate. The dissociation pressure of the air hydrate is given by the dissociation pressure of hydrates formed by the pure gas species multiplied by the mole fraction of the gas species in air (Miller 1969). Accordingly, the air hydrate dissociation pressure in the ice is mainly determined by the dissociation pressure of $N_2$ and $O_2$ with minor contributions from Ar or $CO_2$. We assume that any SII clathrate forming gas species (Sloan and Koh, 2007) will be quantitatively incorporated into the SII air hydrate structure during its nucleation from an air bubble, however we speculate that SI forming gases (such as Xe) may not or not quantitatively be incorporated in the SII air hydrate structure in the ice.

As described in Uchida et al. (2011), the number but also the radius of air hydrate crystals increases in the BCTZ at Dome Fuji, which has very similar climatic conditions compared to EDC. This can be interpreted that in the BCTZ slowly more and more hydrates nucleate from bubbles and that bubbles and clathrates coexist for a long time. Due to the higher partial pressures of gas species in bubbles compared to clathrates a constant permeation flux of gas from the bubbles to the clathrates exists, which is the main cause of clathrate growth in the BCTZ at Dome Fuji (or EDC). Note that due to the different permeation constants of different gas species in ice, this transport implies a fractionation of different gas species which enriches fast permeating gases (such as $O_2$ (Ikeda et al., 2000)) in the hydrate and depletes them in the remaining bubble. For Xe we speculate that there is either no permeation flux (if Xe is not included in the SII hydrate structure) or the permeation flux is very small due to the very low permeation coefficient of the large Xe atom. Thus Xe is depleted in the air hydrate forming in

the BCTZ and enriched in the remaining bubbles. After complete collapse of the bubble, we expect the Xe to be "dissolved" in the ice matrix.

Below the BCTZ Uchida et al. (2011) observe an increase in hydrate radius and at the same time a decline in hydrate number. This can be explained by Ostwald ripening, leading to the growth of large hydrates at the cost of small ones through the Gibbs-Thomson effect (Uchida et al., 2011). Again, the accompanying gas transport from a small to a large hydrate crystal is supplied by (fractionating) permeation of gas species through the ice. If this process is going on for a sufficiently long time, however, a diffusive re-equilibration of the mole fractions of gas species in the hydrates (such as $CO_2$) has been observed (Lüthi et al., 2010). We assume that this permeation acts on all SII forming gas species, but that SI forming Xe is either not permeating at all or that the permeation flux is strongly reduced compared to other gas species. Thus Xe would be depleted in the large clathrates formed by this Ostwald ripening process and would remain "dissolved" in the ice matrix after complete disappearance of a small clathrate.

We attribute the gradually increasing offsets in reconstructed MOT at the top of the BCTZ to core-cracking gas loss in combination with the fractionation of gases by permeation. Gas enclosed in bubbles is more likely to be lost through cracks after core retrieval than gas in hydrates thus, if post-coring gas loss from bubbles occurs, the remaining gas of the ice sample will be enriched in $O_2$ relative to $N_2$ and depleted in Xe. The small but significant $\delta Xe/Kr$ and $\delta Xe/N_2$ decrease in the BCTZ can therefore be explained by the observation that during the gas loss process, Xe is preferentially lost via cracks from the Xe-enriched bubbles.

To the authors: well done! This is a beautiful piece of science and will no doubt have lasting value.
Please also note the supplement to this comment:
https://cp.copernicus.org/preprints/cp-2020-127/cp-2020-127-RC2-supplement.pdf

Here we will not list grammar or typo corrections suggested by referee #2 but we corrected them in the revised version (see also differential Latex file). We will shortly respond to main points made in the annotated manuscript

equation (1): we clarified the units

page 8: we recalculated the values using the local gravitational acceleration

page 9 decrease/increase issue. We apologize that our wording was unclear. We agree with referee #2, but to avoid any confusion, we deleted this sentence as it was not crucial

page 13: we changed the wording as suggested by the referee and added (Jeff Severinghaus, personal communication)

page 20: we revised the discussion on clathrate formation and growth and the accompanying permeation processes as outlined above